# Male cuticular pheromones stimulate removal of the mating plug and promote re-mating through pC1 neurons in *Drosophila* females

**Minsik Yun[1], Do-Hyoung Kim[1], Tal Soo Ha[2], Kang-Min Lee[1], Eungyu Park[1], Markus Knaden[3,4], Bill S Hansson[3,4], Young-Joon Kim[1]\***

[1]School of Life Sciences, Gwangju Institute of Science and Technology (GIST), Gwangju, Republic of Korea; [2]Department of Biomedical Science, College of Natural Science, Daegu University, Gyeongsan, Republic of Korea; [3]Department of Evolutionary Neuroethology, Max Planck Institute for Chemical Ecology, Jena, Germany; [4]Next Generation Insect Chemical Ecology, Max Planck Centre, Max Planck Institute for Chemical Ecology, Jena, Germany

**\*For correspondence:**
kimyj@gist.ac.kr

**Competing interest:** The authors declare that no competing interests exist.

**Abstract** In birds and insects, the female uptakes sperm for a specific duration post-copulation known as the ejaculate holding period (EHP) before expelling unused sperm and the mating plug through sperm ejection. In this study, we found that *Drosophila melanogaster* females shortens the EHP when incubated with males or mated females shortly after the first mating. This phenomenon, which we termed m̲ale-i̲nduced E̲HP s̲hortening (MIES), requires Or47b+ olfactory and ppk23+ gustatory neurons, activated by 2-methyltetracosane and 7-tricosene, respectively. These odorants raise cAMP levels in pC1 neurons, responsible for processing male courtship cues and regulating female mating receptivity. Elevated cAMP levels in pC1 neurons reduce EHP and reinstate their responsiveness to male courtship cues, promoting re-mating with faster sperm ejection. This study established MIES as a genetically tractable model of sexual plasticity with a conserved neural mechanism.

## eLife assessment

This **important** work unravels how female *Drosophila* can assess their social context via chemosensory cues and modulate the sperm storage process after copulation accordingly. A **compelling** set of rigorous experiments uncovers specific pheromones that influence the excitability of the female brain receptivity circuit and their propensity to discard inseminate from a mating. This insight into neuronal mechanisms of sexual behavior plasticity is of general interest to scientists working in the fields of animal behavior, neuroscience, evolution, and sexual selection, as well as insect chemosensation and reproduction.

## Introduction

Sexual plasticity, the ability to modify sexual state or reproductive behavior in response to changing social conditions, is observed in both vertebrates and invertebrates (*Bruce, 1959*; *Koene and Ter Maat, 2007*; *Roberts et al., 2012*; *Steiger et al., 2008*; *Yagound et al., 2012*). In rodents, exposure to unfamiliar males often leads to the sudden termination of pregnancy, known as the Bruce effect. It is induced by male urinary peptides, such as MHC I peptides, activating the vomeronasal organ (*Becker*

*and Hurst, 2008*; *Leinders-Zufall et al., 2004*; *Zipple et al., 2019*). This effect enhances reproductive fitness of both sexes, by eliminating the offspring of competing males and enabling females to select better mates even after conception. Many species also adapt their reproductive behavior in response to the social-sexual context change (SSCC), involving encounters with new sexual partners or competitors. Understanding the neural circuit mechanisms behind female responses to SSCC emerges as a central focus of neuroscience (*Gaspar et al., 2022*; *Kim et al., 2019*; *Liu et al., 2022*; *Wei et al., 2021*).

*Drosophila melanogaster*, the fruit fly, displays various social behaviors like aggregation, aggression, and sexual behavior (*Bartelt et al., 1985*; *Billeter et al., 2006*; *Billeter and Levine, 2015*). Similar to rodents, they primarily use the olfactory system to communicate socially through pheromones (*Kohl et al., 2015*; *Sengupta and Smith, 2014*). Some of these pheromones act as aphrodisiacs, while others regulate aggression or foster aggregation. For instance, 11-*cis*-vaccenyl acetate (cVA) attracts females but repels males and promotes aggregation (*Billeter and Levine, 2015*; *Kurtovic et al., 2007*; *Mane et al., 1983*). 7-Tricosene (7-T), a cuticular hydrocarbon (CHC) present in males, is an aphrodisiac to females and affects social interactions between males (*Grillet et al., 2006*; *Wang et al., 2011*). On the other hand, 7,11-heptacosadiene (7,11-HD), a related female-specific pheromone, functions as an aphrodisiac to males, triggering courtship behavior and involving species recognition (*Antony et al., 1985*; *Toda et al., 2012*).

The fruit fly's chemosensory organs, located in various parts of the body, detect these pheromones (*Ali et al., 2022*; *Joseph and Carlson, 2015*). Olfactory receptor neurons (ORNs) in the sensilla of the antennae and maxillary palp are responsible for the detection of long-range volatile pheromones like cVA, while short-range pheromones like 7-T are sensed by neurons on the forelegs and labellum (*Joseph and Carlson, 2015*; *Kohl et al., 2015*; *Sengupta and Smith, 2014*).

The olfactory receptor Or47b, expressed in ORNs located in the at4 trichoid sensilla on the third antennal segment, is involved in several sociosexual interactions, including male mating success, mate preference, and female aggression toward mating pairs (*Gaspar et al., 2022*; *Kohlmeier et al., 2021*; *Lone et al., 2015*; *Lone and Sharma, 2012*; *Zhuang et al., 2016*). In males, Or47b senses fatty acid methyl esters and fatty acids that affect mating competition and copulation (*Dweck et al., 2015*; *Lin et al., 2016*). While the role of Or47b in female aggression is well established (*Gaspar et al., 2022*), its involvement in female sexual behavior is uncertain. In both sexes, Or47b ORNs project to VA1v glomeruli, where VA1v projection neurons receive their signal and project to the mushroom body calyx and lateral horn. Male Or47b neurons connect to neurons such as aSP5, aSP8, and aSP9, which express a male-specific transcription factor Fru$^M$ (*Yu et al., 2010*).

CHC pheromones, which function as short-range pheromones, are detected primarily by neurons on the forelegs and the labellum that express gustatory receptors, ionotropic receptors, or the ppk/DEG-ENaC family of sodium channels (*Joseph and Carlson, 2015*; *Kohl et al., 2015*; *Sengupta and Smith, 2014*). CHCs like 7-T and 7,11-HD are sensed by *ppk23*-expressing M and F cells in the tarsi (*Liu et al., 2020*). 7-T and cVA are sensed by M cells expressing *ppk23*, whereas 7,11-HD and 7,11-nonacosadiene are sensed by F cells expressing *ppk23*, *ppk25*, and *ppk29*. In males, 7-T or 7,11-HD affects the neuronal activity of the Fru$^M$-expressing P1 neurons (*Inagaki et al., 2014*; *Kohatsu et al., 2011*; *Sato and Yamamoto, 2020*). However, how these CHCs signal in the female brain remains unknown.

Sperm ejection is a process by which females can remove the male ejaculate or the mating plug after copulation. This phenomenon has been observed in several animal species including feral fowl (*Pizzari and Birkhead, 2000*), black-legged kittiwake (*Wagner et al., 2004*), and dunnock (*Davies, 1983*). In the fruit fly, it typically occurs approximately 90 min after mating (*Lee et al., 2015*). This specific interval, referred to as ejaculate holding period (EHP), is thought to affect sperm usage and fecundity (*Lee et al., 2015*; *Manier et al., 2010*). The neurosecretory neurons in the brain pars intercerebralis (PI) that produce diuretic hormone 44 (Dh44), an insect orthologue of the corticotropin-releasing factor, regulate EHP (*Lee et al., 2015*). There is evidence that *Drosophila* females sense the social-sexual context through sperm ejected by other females. For instance, females were likely to lay more eggs when placed on a food patch containing male ejaculate deposited by other females (*Duménil et al., 2016*). However, it remains unknown whether the SSCC influences sperm ejection and EHP.

Female pC1 neurons, which express a specific transcription factor Dsx$^F$, integrate olfactory and auditory cues associated with male courtship (*Lee et al., 2002*; *Zhou et al., 2014*). The pC1 neurons,

their male counterparts (i.e. P1 neurons), and the ventrolateral subdivision of ventromedial hypothalamus (VMHvl) neurons in mice share conserved circuit configurations and demonstrate functional similarity in coordinating social and sexual behaviors (*Anderson, 2016*; *Jiang and Pan, 2022*). There are 14 Dsx-positive pC1 neurons in each hemisphere of the brain, responsive to the male sex-pheromone cVA and courtship songs (*Kim et al., 2024*; *Zhou et al., 2014*). Connectome analyses identified 10 pC1 neurons that fall into five subtypes, with pC1a, b, and c subtypes associated with mating behavior and pC1d and e subtypes associated with aggression (*Chiu et al., 2023*; *Deutsch et al., 2020*; *Han et al., 2022*; *Kim et al., 2024*; *Schretter et al., 2020*; *Wang et al., 2020*). Although direct evidence connecting pC1 neurons to sperm ejection is limited, they are promising candidates for regulating sperm ejection or EHP, because sperm ejection allows females to eliminate the mating plug and male ejaculate, thereby restoring sexual attractiveness (*Laturney and Billeter, 2016*).

In this study, we demonstrated that two male pheromones, 2-methyltetracosane (2MC) and 7-T, significantly reduced the EHP through *Or47b* neurons and *ppk23* neurons, respectively. These pheromone pathways converge on pC1 neurons, where they increase cAMP levels. The elevated cAMP in pC1 neurons resulted in a reduction of the EHP to a degree that was comparable to the effects of the male pheromones. It also enhanced the excitability of pC1 neurons, making them more responsive to

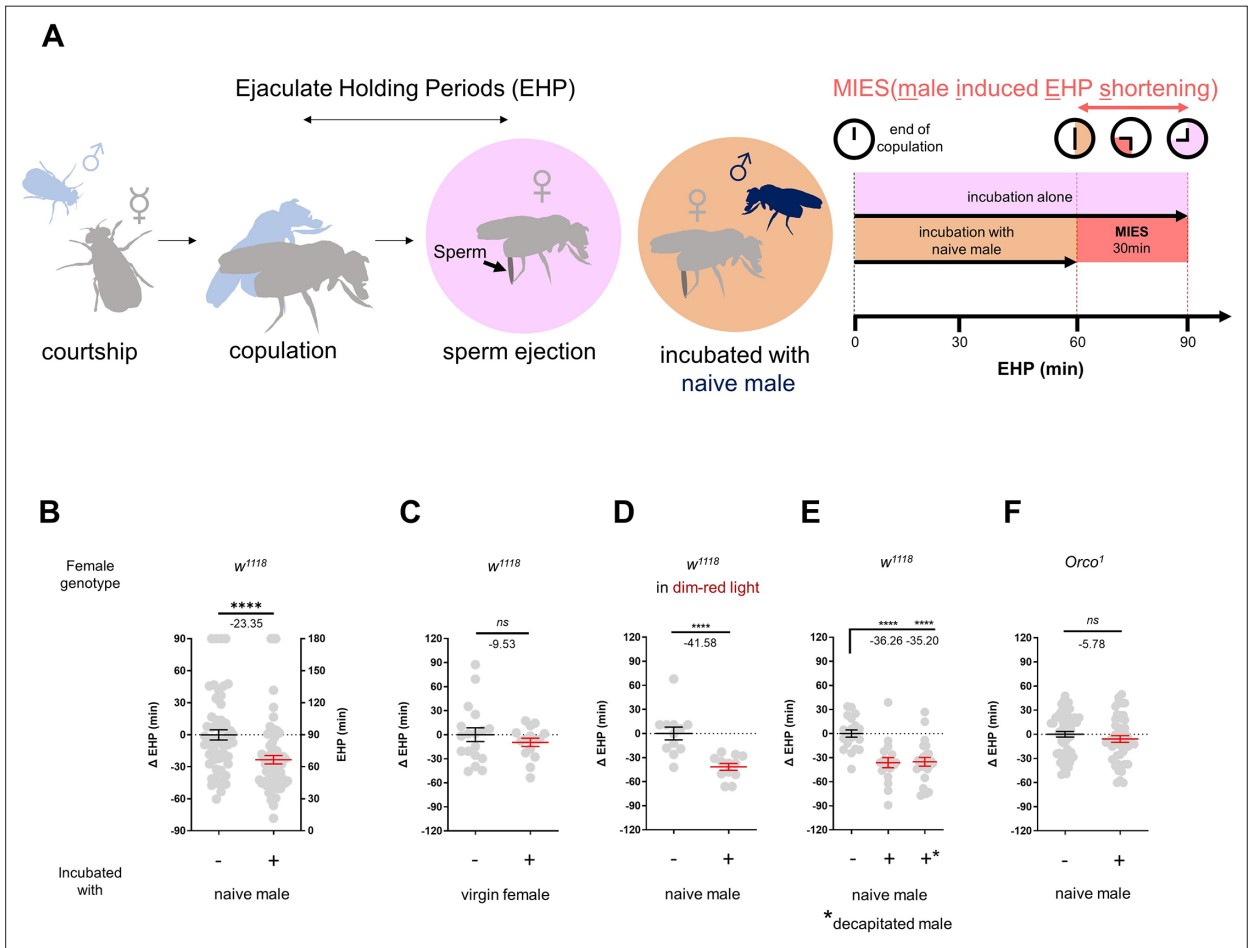

**Figure 1.** The presence of males reduces the ejaculate holding period (EHP) in females through olfactory or gustatory sensation. (**A**) Schematic of the experimental procedure employed to measure male-induced EHP shortening (MIES). Immediately after the end of copulation, the female is incubated with a wild-type *Canton-S* (*CS*) male that has not been previously exposed to the female. Typically, *w^1118* females that are kept alone after mating exhibit an EHP of approximately 90 min, whereas females that are incubated with a naïve *CS* male exhibit an EHP of approximately 60 min. In this study, we refer to this phenomenon as MIES. (**B–F**) Normalized EHP or ΔEHP of the females of the indicated genotypes, incubated under the indicated conditions after mating. The ΔEHP is calculated by subtracting the mean of the reference EHP of females kept alone after mating (the leftmost column) from the EHP of individual females in comparison. Mann-Whitney test (n.s. p>0.05; ****p<0.0001). Gray circles indicate the EHP or ΔEHP of individual females, and the mean ± SEM of data is presented. Numbers below the horizontal bar represent the mean of the EHP differences between the indicated treatments. Genotype and sample size are shown in *Table 1*.

both olfactory and auditory male courtship cues and promoting further mating following the earlier removal of the mating plug. These findings establish a novel behavioral paradigm that sheds light on the intricate molecular and neuronal pathways underlying female sexual plasticity.

## Results

### MIES is dependent on olfaction

To investigate the impact of changes in the social-sexual context on the EHP, we compared the EHP of post-mating females isolated from any male presence to those exposed to naive wild-type *Canton-S* (*CS*) males immediately after copulation (*Figure 1A*). Notably, the EHP of females incubated with naive males was approximately 30 min shorter than that of females left in isolation after mating (*Figure 1A and B*). We refer to this phenomenon as male-induced EHP shortening (MIES). In contrast, little difference in EHP was observed between females incubated with virgin females and those isolated after mating (*Figure 1C*).

Male fruit flies employ various sensory signals to attract females during courtship (*Billeter et al., 2006*). To assess the role of the visual signal in MIES, we examined MIES under dim red light conditions and observed that limited illumination had a marginal impact on MIES (*Figure 1D*). Next, we examined MIES in post-mating females incubated with decapitated *CS* males. These males could serve as a source of olfactory or gustatory signals, but not for auditory or visual signals. Again, no reduction in MIES was observed (*Figure 1E*). This strongly suggests that olfactory or gustatory cues are the key signals responsible for MIES. This is further supported by the observation that females deficient in the odorant receptor co-receptor (*Orco*[1]) did not exhibit MIES (*Figure 1F*). Thus, it is highly likely that male odorant(s), especially those detected by olfactory receptors (Or), induce MIES.

### MIES is dependent on the *Or47b* receptor and *Or47b*-expressing ORNs

In the fruit fly antenna, the trichoid sensilla and their associated ORNs are known to detect sex pheromones (*van der Goes van Naters and Carlson, 2007*). To investigate the contribution of ORNs located in the trichoid sensilla to MIES, we silenced 11 different ORN groups found in the trichoid and intermediate sensilla (*Couto et al., 2005*; *Lin and Potter, 2015*) by expressing either the active or inactive form of tetanus toxin light chain (TNT) (*Sweeney et al., 1995*). Our results showed that silencing ORNs expressing *Or13a, Or19a, Or23a, Or47b, Or65c, Or67d,* or *Or88a* significantly affected MIES (*Figure 2—figure supplement 1*).

We then focused on the analysis of *Or47b*-positive ORNs (*Figure 2A*), which, in contrast to the others, exhibited almost complete abolition of MIES when silenced. Activation of these neurons with the thermogenetic activator dTRPA1 (*Hamada et al., 2008*) resulted in a significant EHP shortening, even in the absence of male exposure (*Figure 2B*). Subsequently, we examined whether restoring Orco expression in *Or47b* ORNs in Orco-deficient females would restore MIES. Our results confirmed that this is indeed the case (*Figure 2C*). To establish the necessity of the *Or47b* receptor gene for MIES, we examined Or47b-deficient females (*Or47b*[2]/*Or47b*[3]) and observed a complete absence of MIES, whereas heterozygous controls exhibited normal MIES (*Figure 2D*). Furthermore, the reintroduction of *Or47b* expression in *Or47b* ORNs of *Or47b*-deficient females almost completely restored MIES (*Figure 2E*). Based on these observations, we concluded that MIES depends on the *Or47b* receptor gene and Or47b-expressing ORNs.

### 2MC induces MIES via *Or47b* and *Or47b* ORNs

Previous studies have shown that methyl laurate and palmitoleic acid can activate *Or47b* ORNs only in the presence of a functional *Or47b* gene (*Dweck et al., 2015*; *Lin et al., 2016*). However, in our investigation, none of these odorants induced significant EHP shortening, even when applied at concentrations as high as 1440 ng (*Figure 3—figure supplement 1*). This prompted us to search for a new pheromone capable of activating Or47b ORNs and thereby shortening the EHP.

Oenocytes produce a significant portion of the CHCs or pheromones. We asked whether the male pheromone responsible for MIES is produced by oenocytes (*Figure 3—figure supplement 2A*). Indeed, incubation with females engineered to produce male oenocytes significantly shorten EHP, strongly suggesting that male oenocytes serve as a source for the MIES pheromone. Unexpectedly, however, incubation with males possessing feminized oenocytes also resulted in significant EHP

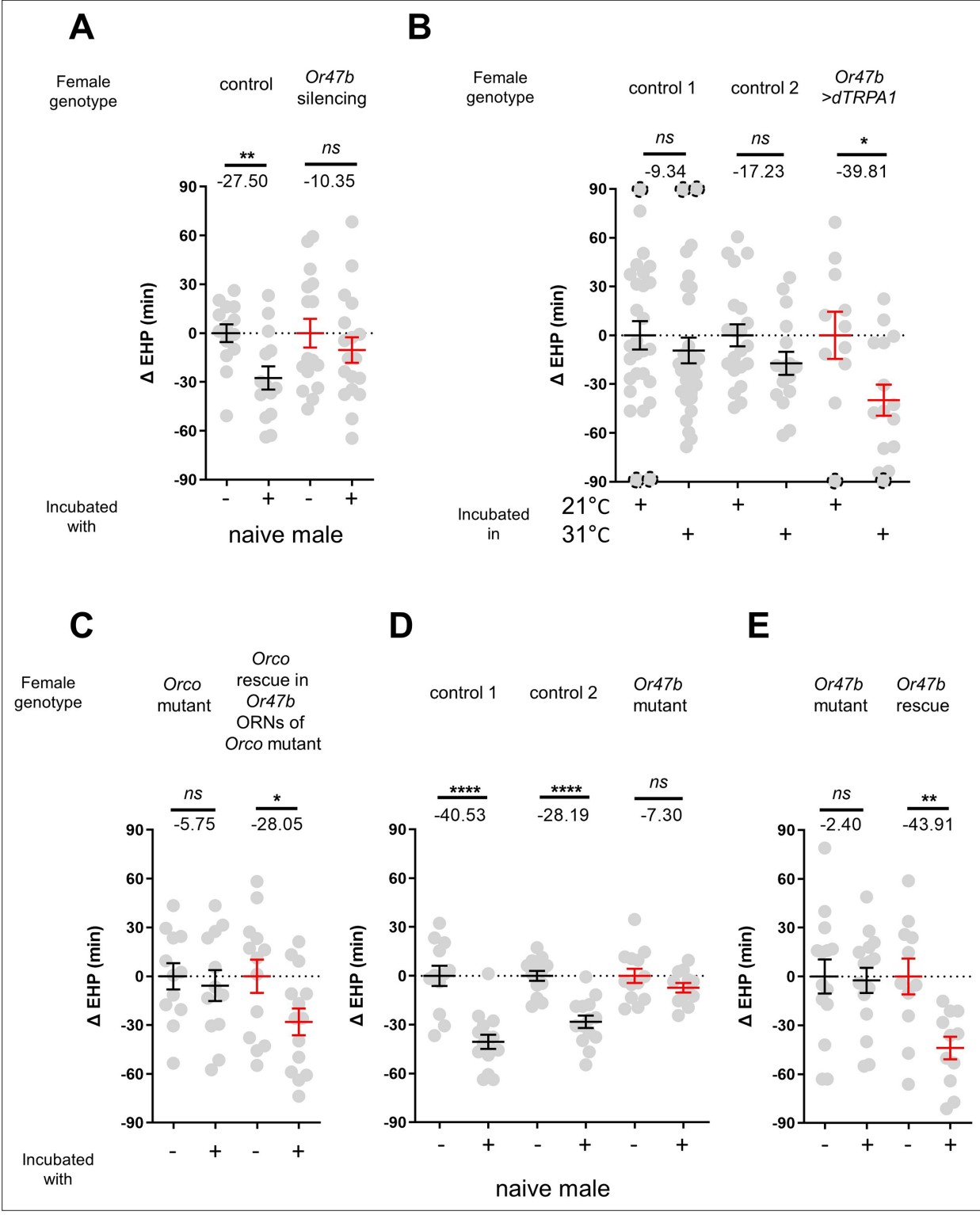

**Figure 2.** The function of Or47b and Or47b-positive olfactory receptor neurons (ORNs) is essential for male-induced EHP shortening (MIES). (**A, C–E**) ΔEHP of females of the indicated genotypes, incubated with or without naive males after mating. The female genotypes are as follows from left to right: (**A**) control (*Or47b>TNT^inactive^*), *Or47b* ORN silencing (*Or47b>TNT^active^*); (**C**) *Orco* mutant (*Orco^1^/Orco^1^*), *Orco* rescue in *Or47b* ORNs of *Orco* mutant (*Orco^1^/Orco^1^; Or47b>Orco*); (**D**) control 1 (*Or47b^2^/+*), control 2 (*Or47b^3^/+*), *Or47b* mutant (*Or47b^2^/Or47b^3^*); (**E**) *Or47b* mutant (*Or47b^2^/Or47b^2^*), *Or47b* rescue (*Or47b>Or47b; Or47b^2^/Or47b^2^*). (**B**) Thermogenetic activation of *Or47b*-positive ORNs shortens EHP in females kept alone after mating. The female genotypes are as follows from left to right: control 1 (*Or47b-Gal4/+*), control 2 (*UAS-dTRPA1/+*), *Or47b>dTRPA1* (*Or47b-Gal4/UAS-dTRPA1*). Mann-Whitney test (n.s. p>0.05; *p<0.05; **p<0.01; ****p<0.0001). The ΔEHP is calculated by subtracting the mean of the reference EHP of females kept

*Figure 2 continued on next page*

*Figure 2 continued*

alone after mating ('-' in **A, C–E**) or incubated at 21°C control conditions (**B**) from the EHP of individual females in comparison. Gray circles indicate the ΔEHP of individual females, and the mean ± SEM of data is presented. The gray circles with dashed borders indicate ΔEHP values that exceed the axis limits (>90 or <-90 min). Numbers below the horizontal bar represent the mean of the EHP differences between the indicated treatments. EHP, ejaculate holding period. Genotype and sample size are shown in *Table 1*.

The online version of this article includes the following figure supplement(s) for figure 2:

**Figure supplement 1.** The identification of trichoid and intermediate sensilla olfactory receptor neurons (ORNs) that are necessary for the production of male-induced EHP shortening (MIES).

shortening (*Figure 3—figure supplement 2A*). This raises the possibility that oenocytes may not be the sole source of the MIES. The genus *Drosophila* exhibits distinct CHC profiles, with certain CHC components shared among closely related species (*Billeter et al., 2009*). We found that incubation with males of other closely related species, such as *Drosophila simulans*, *Drosophila sechellia*, and *Drosophila erecta,* also induced EHP shortening, whereas incubation with *Drosophila yakuba* males did not (*Figure 3—figure supplement 2B*).

The EHP was therefore measured in females incubated in a small mating chamber containing a piece of filter paper perfumed with male CHCs, including 2-methylhexacosane, 2-methyldocosane,

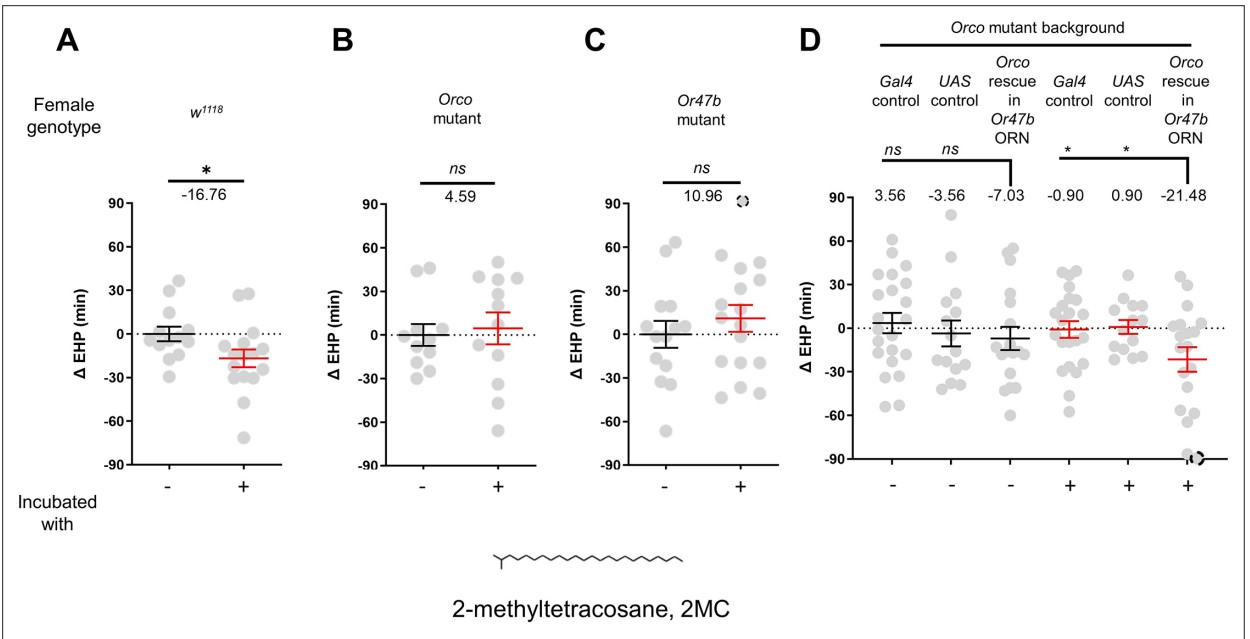

**Figure 3.** 2-Methyltetracosane (2MC) can induce ejaculate holding period (EHP) shortening through Or47b. (**A–D**) ΔEHP of females of the indicated genotypes, incubated in solvent vehicle or 2MC. Mated females were incubated with a piece of filter paper perfumed with either vehicle (-) or 750 ng 2MC (+). The female genotypes are as follows: (**A**) $w^{1118}$, (**B**) *Orco* mutant (*Orco$^1$/Orco$^1$*), (**C**) *Or47b* mutant (*Or47b$^2$/Or47b$^2$*), (**D**) *Gal4* control (*Or47b-Gal4/+; Orco$^1$/Orco$^1$*), *UAS* control (*UAS-Orco/+; Orco$^1$/Orco$^1$*), *Orco* rescue in *Or47b* olfactory receptor neurons (ORN) (*Orco$^1$/Orco$^1$; Or47b-Gal4/ UAS-Orco*). (**A–C**) Mann-Whitney test (n.s. p>0.05; *p<0.05). (**D**) One-way analysis of variance (ANOVA) test with Fisher's LSD multiple comparison (n.s. p>0.05; *p<0.05). Gray circles indicate the ΔEHP of individual females and the mean ± SEM of data is presented. The ΔEHP is calculated by subtracting the mean of the reference EHP of females incubated with vehicle-perfumed paper (the leftmost column in **A–C**) or the mean of the *Gal4* control and *UAS* control female incubated with vehicle-perfumed paper (the two leftmost columns in **D**) from the EHP of individual females in comparison. Gray circles with dashed borders indicate ΔEHP values that exceed the axis limits (>90 or <-90 min). Numbers below the horizontal bar represent the mean of the EHP differences between the indicated treatments. Genotype and sample size are shown in *Table 1*.

The online version of this article includes the following figure supplement(s) for figure 3:

**Figure supplement 1.** Known odorant ligands for *Or47b*, methyl laurate and *trans*-palmitoleic acid, were unable to induce ejaculate holding period (EHP) shortening.

**Figure supplement 2.** Ejaculate holding period (EHP) shortening is induced by males with feminized oenocytes, females with masculinized oenocytes, and males of other closely related *Drosophila* species.

**Figure supplement 3.** 2-Methyltetracosane (2MC) shortens ejaculate holding period (EHP) at a specific concentration.

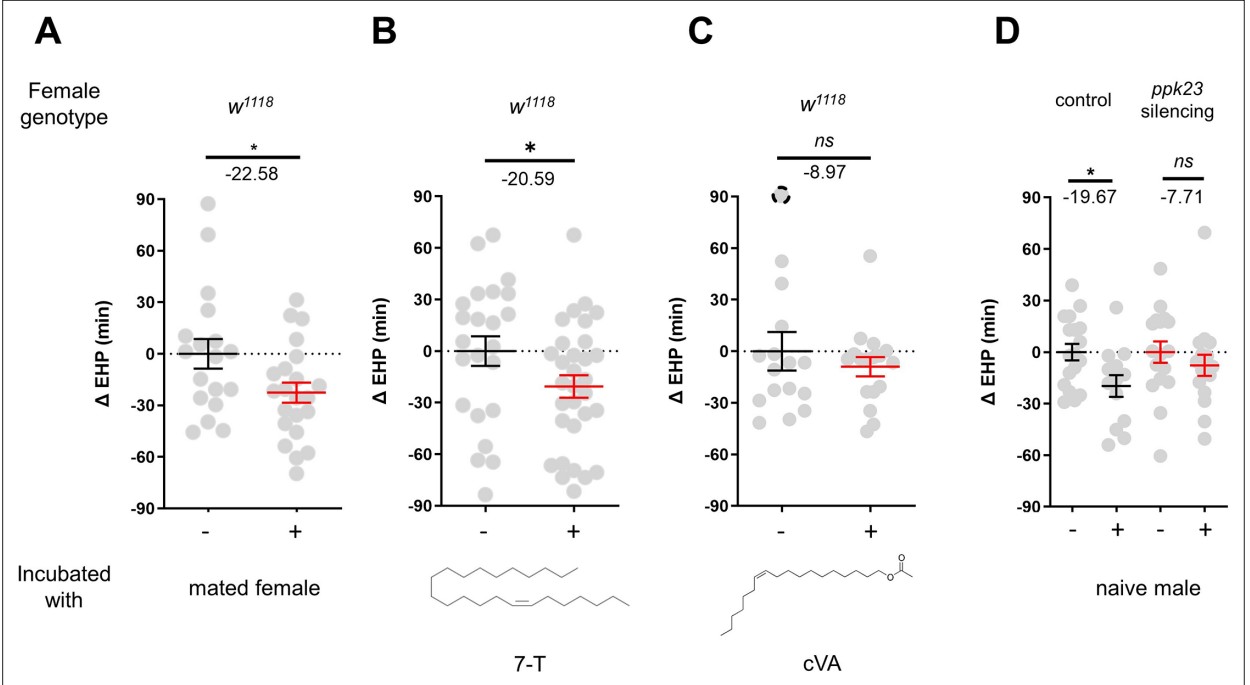

**Figure 4.** 7-Tricosene (7-T) present in mated females and males reduces ejaculate holding period (EHP) via *ppk23* neurons. (**A–D**) ΔEHP of females of the indicated genotypes, incubated with mated females (**A**), a piece of filter paper perfumed with 150 ng 7-T (**B**), 200 ng 11-*cis*-vaccenyl acetate (cVA) (**C**), or naive males (**D**) after mating. The female genotypes are as follows: (**A–C**) *w1118*, (**D**) control (*ppk23-Gal4/UAS-TNTinactive*), *ppk23* silencing (*ppk23-Gal4/UAS-TNTactive*). (**A**) Unpaired t-test. (**B–D**) Mann-Whitney test (n.s. p>0.05; *p<0.05). The ΔEHP is calculated by subtracting the mean of the reference EHP of females kept alone ('-' in **A, D**) or incubated with vehicle-perfumed paper (the leftmost column in **B, C**) from the EHP of individual females in comparison. Gray circles indicate the ΔEHP of individual females, and the mean ± SEM of data is presented. The gray circles with dashed borders indicate ΔEHP values that exceed the axis limits (>90 or <-90 min). Numbers below the horizontal bar represent the mean of the EHP differences between the indicated treatments. Genotype and sample size are shown in **Table 1**.

The online version of this article includes the following figure supplement(s) for figure 4:

**Figure supplement 1.** 7-Tricosene (7-T) induces ejaculate holding period (EHP) shortening at physiological concentrations, but DEG/ENaC channels expressed in *ppk23* neurons are not required for male-induced EHP shortening (MIES).

5-methyltricosane, 7-methyltricosane, 10Z-heneicosene, 9Z-heneicosene, and 2MC at various concentrations (not shown). Among these, 2MC at 750 ng was the only one that significantly reduced EHP (**Figure 3A**; **Figure 3—figure supplement 3**). 2MC was mainly found in males, but not in virgin females (**Dweck et al., 2015**). Notably, it is present in *D. melanogaster*, *D. simulans*, *D. sechellia*, and *D. erecta*, but not in *D. yakuba* (**Dweck et al., 2015**; **Wang et al., 2022**).

Moreover, the 2MC-induced EHP shortening was not observed in Orco- or Or47b-deficient females (**Figure 3B and C**), but was restored when Orco expression was reinstated in *Or47b* ORNs in Orco-deficient mutants (**Figure 3D**). Our behavioral observations strongly suggest that 2MC acts as an odorant ligand for Or47b and shortens the EHP through this receptor.

## 7-T shortens EHP through *ppk23* neurons

In contrast to incubation with virgin females, incubation with mated females resulted in a significant shortening of EHP (**Figure 4A**). Mated females carry male pheromones, including 7-T and cVA, which are transferred during copulation (**Laturney and Billeter, 2016**). This raised the possibility that these male pheromones might also induce EHP shortening. Indeed, our experiments revealed that incubation with a piece of filter paper perfumed with 150 ng of 7-T significantly shortened the EHP. Conversely, incubation with cVA and 7-pentacosene, a related CHC, did not produce the same effect (**Figure 4B and C**; **Figure 4—figure supplement 1A, B**). The concentrations of 7-T capable of inducing EHP shortening appear to be physiologically relevant. 7-T has been found at levels of 432 ng in males (**Scott and Richmond, 1988**), 25 ng in virgin females, and 150 ng in mated females (**Laturney and Billeter, 2016**). Although the receptors for 7-T remain unknown, *ppk23*-expressing tarsal neurons

have been shown to sense these compounds and regulate sexual behavior in males and females (*Thistle et al., 2012*; *Toda et al., 2012*; *Vijayan et al., 2014*). Subsequently, we silenced *ppk23* neurons, and as a result, MIES was almost completely abolished, underscoring the pivotal role of 7-T in MIES (*Figure 4D*). However, DEG/ENac channel genes expressed in *ppk23* neurons, including *ppk23* and *ppk29*, were found to be dispensable for MIES (*Figure 4—figure supplement 1C–E*). This aligns with the previous observations that neither *ppk23* deficiency nor *ppk28* deficiency recapitulates the sexual behavioral defects caused by silencing *ppk23* neurons (*Lu et al., 2012*).

## The pC1b and c neurons regulate EHP and MIES

The neuropeptide Dh44 determines the timing of sperm ejection or EHP (*Lee et al., 2015*). The same study found that Dh44 receptor neurons involved in EHP regulation also express *doublesex* (*dsx*), which encodes sexually dimorphic transcription factors. A recent study has revealed that pC1 neurons, a specific subgroup of *dsx*-expressing central neurons in the female brain, do indeed express Dh44 receptors (*Kim et al., 2024*). With these findings, we set out to investigate the role of pC1 neurons in the regulation of EHP and MIES. The pC1 neurons comprise five distinct subtypes. Of these, the pC1a, b, and c subtypes have been implicated in mating receptivity (*Deutsch et al., 2020*; *Kim et al., 2024*), while the remaining pC1d and e subtypes have been implicated in female aggression (*Chiu et al., 2023*; *Deutsch et al., 2020*). To investigate the role of these subtypes in EHP, we employed GtACR1, an anion channel activated by blue light in the presence of all *trans*-retinal (ATR), to silence specific pC1 subtypes immediately after mating. Our experiments revealed that silencing of the pC1 subset comprising the pC1a, b, and c subtypes with GtACR1 led to an increase in EHP (*Figure 5A*), whereas silencing of the pC1d and e subtypes had a limited effect on EHP (*Figure 5B*). We further analyzed the roles of pC1b, c neurons along with pC1a neurons separately. We generated a subtype-specific split-Gal4 for pC1a and found that, as expected, silencing pC1a with this split-Gal4 almost completely suppressed mating receptivity (*Figure 5—figure supplement 1*). However, silencing pC1a alone did not result in increased EHP, suggesting a marginal role of the pC1a subtype in EHP regulation (*Figure 5C*). In contrast, concomitant silencing of both pC1b and pC1c neurons significantly increased EHP by 56±6.9 min (*Figure 5D*). At present, we lack the genetic tools to further distinguish the roles of pC1b and pC1c subtypes separately.

## 2MC and 7-T increase cAMP levels in pC1b and c neurons

Our recent research has shown that pC1 neurons exhibit elevated cAMP levels during sexual maturation, and that this increase in cAMP is closely related to heightened excitability of pC1 neurons (*Kim et al., 2024*). The same study also showed that a mating signal (i.e. sex peptide in the male seminal fluid) reduces cAMP levels in pC1 neurons. Thus, we hypothesized that male odorants responsible for inducing MIES, such as 2MC or 7-T, would elevate cAMP levels in pC1b, c neurons in newly mated females. This, in turn, would lead to increased excitability of pC1 neurons and, as a consequence, a reduction in the EHP. To monitor cAMP levels in these neurons, we prepared females that express a CRE-luciferase reporter selectively in pC1b, c neurons. Indeed, when exposed to 2MC or 7-T, pC1b, c neurons exhibited a significant increase in CRE-luciferase activity, indicating that these neurons produce higher levels of cAMP in response to these odorants (*Figure 5E*). Notably, CRE-luciferase activity appeared to peak at specific odorant concentrations that induced significant shortening of the EHP (*Figure 5—figure supplement 2*).

In contrast, when we examined other pC1 subsets, such as pC1a, and pC1d and e, we detected no evidence of increased CRE-luciferase reporter activity upon exposure to 2MC or 7-T treatment (*Figure 5E*). Notably, CRE-luciferase reporter activity in the pC1a neurons appears to be dependent on the mating status, as it reaches levels similar to those of pC1b, c neurons in virgin females (*Figure 5—figure supplement 3*). This observation aligns well with connectome data suggesting that SAG neurons, which are responsible for relaying SP-dependent mating signals, synapse primarily with the pC1a subtype and to a much lesser extent with other pC1 subtypes (*Wang et al., 2020*).

To further test the role of Or47b in 2MC detection, we generated Or47b-deficient females with pC1 neurons expressing the CRE-luciferase reporter. Females with one copy of the wild-type Or47b allele, which served as the control group, showed robust CRE-luciferase reporter activity in response to either 2MC or 7-T. In contrast, *Or47b*-deficient females showed robust CRE-luciferase activity in

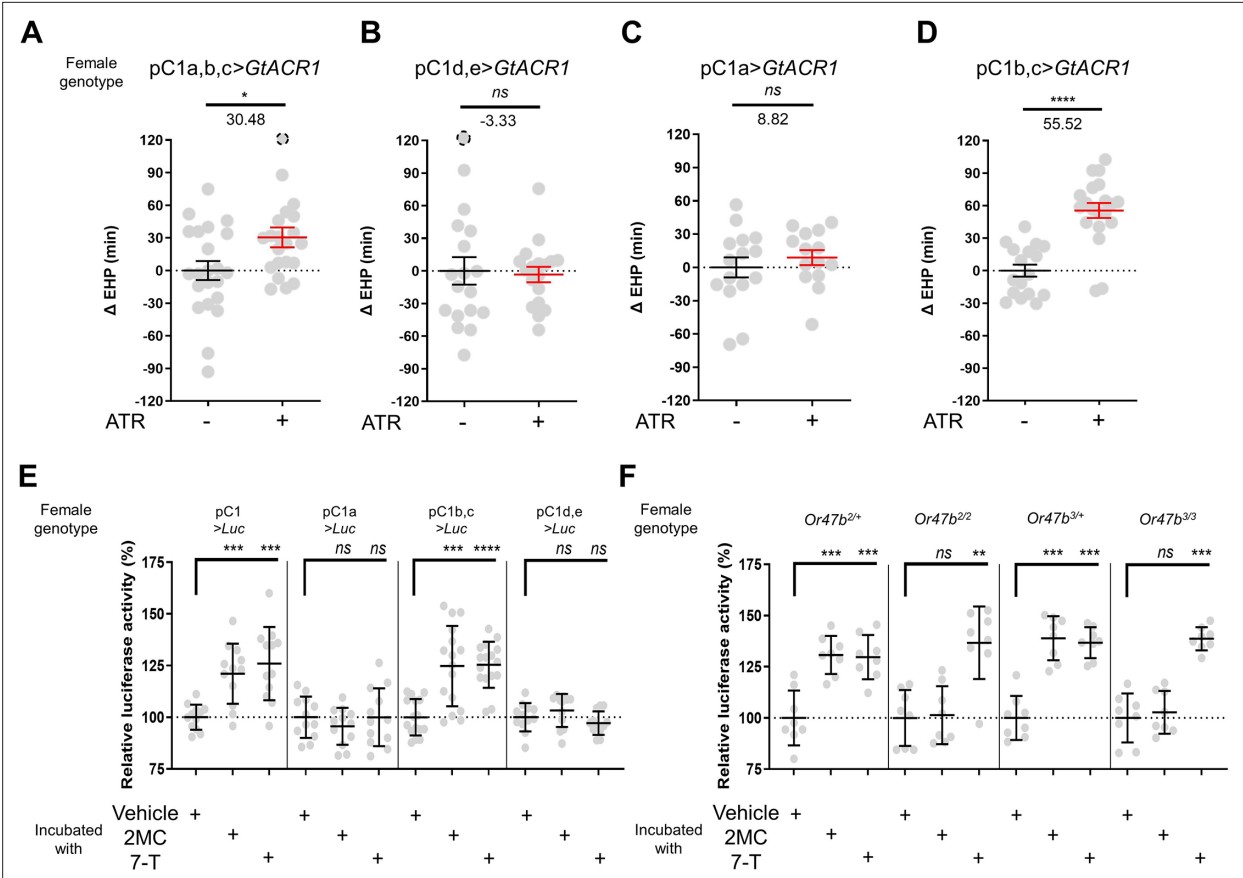

**Figure 5.** A subset of pC1 neurons, comprising pC1b and pC1c subtypes, regulates ejaculate holding period (EHP) and exhibits CRE-luciferase reporter activity in response to 2-methyltetracosane (2MC) and 7-tricosene (7-T). (**A–D**) The optogenetic silencing of a pC1 neuron subset comprising pC1b and pC1c neurons (pC1b, c) increases EHP. Females of the indicated genotypes were cultured on food with or without all *trans*-retinal (ATR) after eclosion. The ΔEHP is calculated by subtracting the mean of the reference EHP of females cultured in control ATR - food (the leftmost column) from the EHP of individual females in comparison. The female genotypes are as follows: (**A**) pC1a,b,c>*GtACR1* (*pC1-S-Gal4/UAS-GtACR1*), (**B**) pC1d,e>*GtACR1* (*pC1-A-Gal4/UAS-GtACR1*), (**C**) pC1a>*GtACR1* (*pC1a-split-Gal4/UAS-GtACR1*), and (**D**) pC1b,c>*GtACR1* (*Dh44-pC1-Gal4/UAS-GtACR1*). Gray circles indicate the ΔEHP of individual females, and the mean ± SEM of data is presented. The gray circles with dashed borders indicate ΔEHP values that exceed the axis limits (>120 min). Mann-Whitney test (n.s. p>0.05; *p<0.05; ****p<0.0001). Numbers below the horizontal bar represent the mean of the EHP differences between the indicated treatments. (**E, F**) Relative CRE-luciferase reporter activity of pC1 neurons in females of the indicated genotypes, incubated with a piece of filter paper perfumed with solvent vehicle control or the indicated pheromones immediately after mating. The CRE-luciferase reporter activity of pC1 neurons of Or47b-deficient females (*Or47b²/²* or *Or47b³/³*) was observed to increase in response to 7-T but not to 2MC. To calculate the relative luciferase activity, the average luminescence unit values of the female incubated with the vehicle are set to 100%. Mann-Whitney test (n.s. p>0.05; **p<0.01; ***p<0.001; ****p<0.0001). Gray circles indicate the relative luciferase activity (%) of individual females, and the mean ± SEM of data is presented. Genotype and sample size are shown in **Table 1**.

The online version of this article includes the following source data and figure supplement(s) for figure 5:

**Figure supplement 1.** Characterization of *pC1a-split-Gal4*.

**Figure supplement 1—source data 1.** Raw image file for the confocal Z-projection image of *pC1a-split-GAL4* neurons in a female brain.

**Figure supplement 2.** Incubation with 2-methyltetracosane (2MC) or 7-tricosene (7-T) increases cAMP levels in pC1 neurons.

**Figure supplement 3.** Incubation with 2-methyltetracosane (2MC) or 7-tricosene (7-T) increases cAMP levels in pC1a as well as pC1b, c neurons in virgin females.

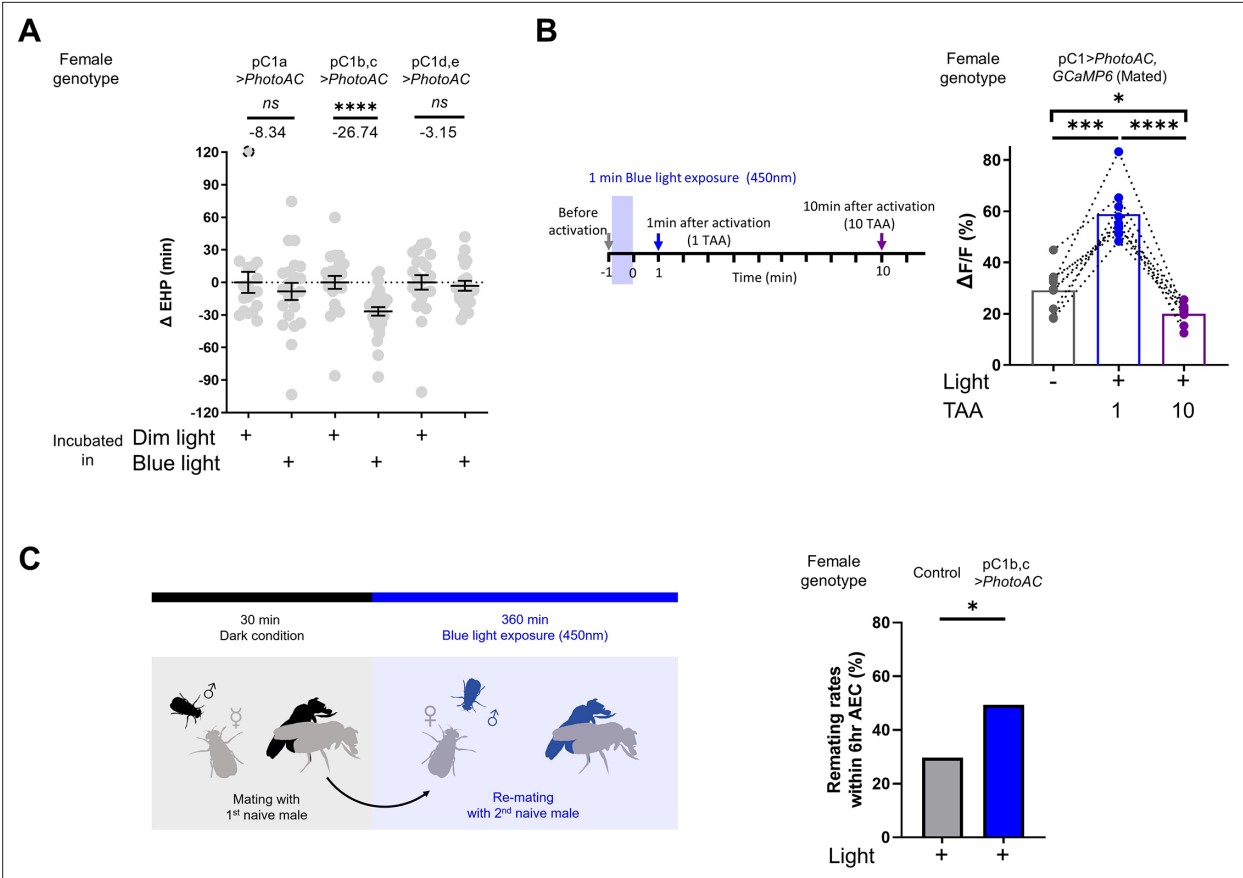

**Figure 6.** Elevated cAMP levels in pC1 neurons reduce ejaculate holding period (EHP) and increase the responsiveness of pC1 neurons to male courtship cues, thereby promoting subsequent re-mating. (**A**) The optogenetic production of cAMP in the pC1b, c neurons shortens EHP, whereas the same treatment in pC1a or pC1d, e neurons does not. ΔEHP is calculated by subtracting the mean of the reference EHP of females incubated in the control illumination (Dim light), which does not activate a photoactivatable adenylate cyclase (PhotoAC), from the EHP of individual females. Mann-Whitney test (n.s. p>0.05, ****p<0.0001). (**B**) The optogenetic production of cAMP transiently increases the excitability of pC1 neurons. Left, schematic of the experimental procedure. Right, peak ΔF/F in the LPC projections of pC1 neurons from freshly mated females in response to the pheromone 11-*cis*-vaccenyl acetate (cVA), before and after photoactivation of PhotoAC expressed in pC1 neurons. The calcium response was measured at specific time points after photoactivation: after 1 min (blue dots and box) or 10 min (purple dots and box) after activation. Repeated measures one-way ANOVA test with the Geisser-Greenhouse correction followed by Tukey's multiple comparisons test (*p<0.05; ***p<0.001; ****p<0.0001). (**C**) Left, schematic of the experimental procedure. Right, re-mating rate of females during optogenetic cAMP production in pC1b, c neurons, scored as the percentage of females that copulate with a naive *Canton-S* (*CS*) male within 6 hr after the first mating. The female genotypes are as follows: Control (+/*UAS-PhotoAC*), pC1b,c>*UAS-PhotoAC* (*Dh44-pC1-Gal4/UAS-PhotoAC*). Chi-square test (*p<0.05). Genotype and sample size are shown in *Table 1*.

The online version of this article includes the following figure supplement(s) for figure 6:

**Figure supplement 1.** The knockdown of Dh44R1 and Dh44R2 in pC1 neurons has a limited impact on male-induced EHP shortening (MIES).

response to to 7-T, but little activity in response to 2MC. This observation suggests that the odorant receptor Or47b plays an essential role in the selective detection of 2MC (*Figure 5F*).

## Elevated cAMP in pC1 neurons shortens the EHP, while increasing re-mating

Having shown that MIES-inducing male odorants, 2MC or 7-T, increase cAMP levels in pC1b, c neurons from mated females, we next asked whether this induced elevation of cAMP levels in pC1b, c neurons would shorten EHP, leading to MIES. We employed the photoactivatable adenylate cyclase (PhotoAC), which increases cellular cAMP levels upon exposure to light. Indeed, the induced elevation of cAMP levels in pC1b, c neurons significantly shortened EHP, whereas the same treatment applied to pC1a or pC1d and pC1e had no such effect (*Figure 6A*). This further underscores the pivotal role of pC1b, c neurons in EHP regulation.

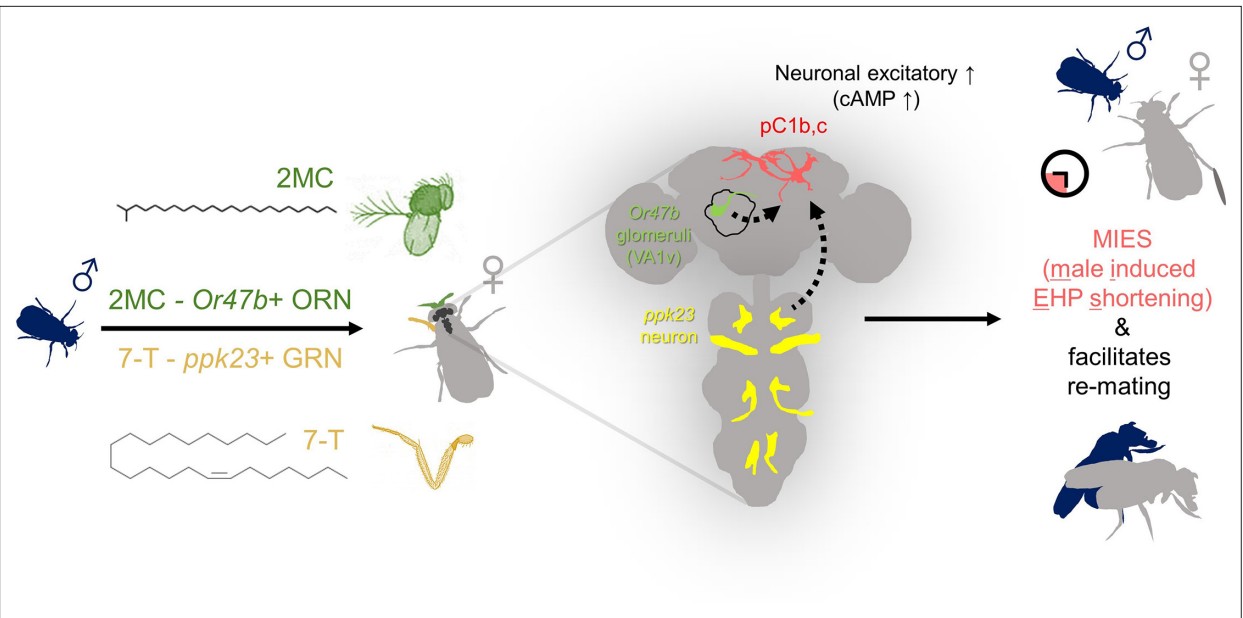

**Figure 7.** The presence of male odorants, which reflect changes in the social-sexual context, stimulates newly mated females to remove the male ejaculate and engage in subsequent re-mating. Following the initial mating, a female that encounters a new courting male removes the male ejaculate after a shorter ejaculate holding period (EHP) than those that do not encounter new male partners. This phenomenon, referred to as male-induced EHP shortening (MIES) in this study, is followed by a second mating with the new partner. The production of MIES depends on the functions of the Or47b+ olfactory and ppk23+ gustatory neurons, which are activated by 2-methyltetracosane (2MC) and 7-T, respectively. These odorants increase cAMP levels in pC1b, c neurons, enhancing their responsiveness to male courtship cues and increasing mating receptivity. Consequently, 2MC and 7-T promote a second mating with a faster removal of the male ejaculate or mating plug.

Next, we asked whether the expression of Dh44R1 and Dh44R2, GPCRs that increase cellular cAMP in response to their ligand Dh44, in pC1b, c neurons is necessary for MIES. However, double knockdown of Dh44R1 and Dh44R2 in pC1 neurons seemed to have a limited impact on MIES (*Figure 6— figure supplement 1*). This suggests that Dh44R signaling in pC1 neurons is not essential for the regulation of EHP or MIES, raising the possibility that other GPCRs may be involved in the upregulation of cAMP levels in pC1 neurons in response to 2MC or 7-T.

Lastly, we investigated how increased cAMP levels affect the physiological activity of pC1 neurons. pC1 neurons from virgin females exhibit robust $Ca^{2+}$ transients in response to male courtship cues, such as the male pheromone cVA and the courtship pulse song (*Zhou et al., 2014*). In contrast, those from mated females display significantly diminished $Ca^{2+}$ transients (*Kim et al., 2024*). When examined shortly after mating, a decrease in pC1 responsiveness to cVA was observed. However, immediately after PhotoAC activation in pC1 neurons, pC1 neurons from freshly mated females became more excitable and exhibited stronger $Ca^{2+}$ transients in response to cVA (*Figure 6B*). It is important to note that this PhotoAC-induced increase in pC1 excitability is transient and rapidly declines within 10 min (*Figure 6B*). Nevertheless, these findings suggest that the increased cAMP levels in pC1 neurons would not only promote MIES but also facilitate re-mating in post-mating females, which typically engage in re-mating at a low frequency. To test this hypothesis, we examined the re-mating frequency of freshly mated females paired with naive males while inducing a cAMP increase in pC1 neurons. As expected, PhotoAC activation in pC1b, c neurons substantially increased the re-mating rate compared to the control group (*Figure 6C*). Therefore, we concluded that male odorants that stimulate cAMP elevation in pC1 neurons expedite the removal of the mating plug, consequently leading to increased instances of re-mating (*Figure 7*).

## Discussion

Males employ a diverse range of strategies to enhance their reproductive fitness. One such strategy involves the formation of a 'mating plug', a mechanism that prevents females from engaging in further mating, thereby increasing fertilization success rates (*Dixson, 1998*; *Parker, 1970*; *Schneider et al.,*

*2016*). As a means of intra-sexual competition, rival males often promote the removal or precocious expulsion of the mating plug. The evolution of this strategy is driven by intersexual interactions with polyandrous females, who often remove the mating plug to engage in additional mating with males of superior traits or higher social status than their previous partners (*Dean et al., 2011*; *Pizzari and Birkhead, 2000*). In the dunnock *Prunella modularis*, a small European passerine bird, the male often engages in cloacal pecking of mated females, inducing the expulsion of the previous mate's sperm and mating plug, thereby increasing their chances of successful mating (*Davies, 1983*). In this study, we discovered that in *D. melanogaster*, freshly mated females exhibit an earlier removal of the mating plugs or a shorter EHP when kept with actively courting males. This behavior is primarily induced by the stimulation of females via male sex pheromones. In addition, our study has revealed that the neural circuit that processes male courtship cues and controls mating decisions plays an important role in regulating this behavior. This fly circuit has recently been proposed to be homologous to VMHvl in the mouse brain (*Anderson, 2016*; *Jiang and Pan, 2022*). By delving into the molecular and neuronal mechanisms underlying MIES, our study provides valuable insights into the broader aspect of behaviors induced by changes in the social-sexual context.

Our findings highlight the involvement of the Or47b receptor and Or47b ORNs in MIES. These OR and ORNs have been implicated in a range of social and sexual behaviors in both male and female fruit flies (*Dweck et al., 2015*; *Gaspar et al., 2022*; *Kohlmeier et al., 2021*; *Lin et al., 2016*; *Lone et al., 2015*; *Lone and Sharma, 2012*; *Zhuang et al., 2016*). Methyl laurate and *trans*-palmitoleic acid are odorant ligands for Or47b that account for many of these functions, particularly in males (*Dweck et al., 2015*; *Lin et al., 2016*). In this study, we provide compelling evidence that 2MC induces cAMP elevation in pC1 neurons and EHP shortening via both the Or47b receptor and Or47b ORNs, suggesting that 2MC functions as an odorant ligand for Or47b. Notably, gas chromatography-mass spectrometry analysis of CHCs from 4-day-old wild-type *D. melanogaster* revealed the presence of 2MC exclusively in males (*Dweck et al., 2015*). Surprisingly, however, unlike 2MC, neither methyl laurate nor *trans*-palmitoleic acid affected EHP. The reason for this paradoxical result remains unclear. A plausible interpretation is that the EHP shortening induced by 2MC may require not only Or47b but also other as yet unidentified ORs. With the establishment of a behavioral and cellular assessment of 2MC activity, the search for additional odorant receptors responsive to 2MC is now feasible. Another important avenue for further research is whether 2MC can also elicit behaviors previously associated with methyl laurate or *trans*-palmitoleic acid, such as promoting male copulation and courtship (*Dweck et al., 2015*; *Lin et al., 2016*).

We observed that both 2MC and 7-T exhibit both cellular and behavioral activity within a specific concentration range (*Figure 3—figure supplement 3*, *Figure 4—figure supplement 1*, *Figure 5—figure supplement 2*). This observation is of particular interest, given the multitude of environmental and biological factors that influence the levels of 2MC and 7-T, potentially affecting the capacity of males to induce MIES. For instance, exposure to low temperatures during development has been linked to increased production of both 2MC and 7-T (*Bontonou et al., 2013*). Similarly, mutation of the desiccation stress gene CG9186, which encodes a protein associated with lipid droplets, has been found to impact 2MC levels (*Werthebach et al., 2019*). Furthermore, 2MC levels rise with age in males (*Everaerts et al., 2010*). Thus, we propose that levels of 2MC, and possibly 7-T, may serve as indicators of male age and resilience to environmental stress in a complex manner.

In mated females, treatment with 2MC or 7-T increases cAMP levels in pC1b, c neurons but not in pC1a neurons. In contrast, pC1a neurons in virgin females are fully responsive to both male pheromones, showing increases in cAMP levels that are similar to those of pC1b,c neurons (*Figure 5—figure supplement 3*). The absence of cAMP levels in pC1a neurons in mated females likely results from the mating signal (i.e. sex peptide) silencing pC1a neurons. Connectome and electrophysiology data support this interpretation, as SAG neurons, which relay sex peptide signals, have the strongest synaptic connection with the pC1a among five pC1 subtypes (*Wang et al., 2020*). However, the activity of SAG neurons may also influence pC1c neurons, as they also have substantial synaptic connections with pC1c neurons, as seen in the hemibrain connectome dataset (*Scheffer et al., 2020*). Future studies are needed to understand the role of SAG neurons in the regulation of EHP.

We found that increased cAMP levels cause pC1b, c neurons in mated females, which are typically unresponsive to male courtship cues like cVA and pulse song, to become responsive and exhibit strong Ca²⁺ transients. Since pC1b, c neurons play a role in generating sexual drive and increasing

female receptivity to male courtship, the 2MC- or 7-T-induced increases in cAMP are likely to control the removal of the mating plug and the engagement of mated females in further mating (*Figure 7*). This hypothesis aligns well with the previous report that mating reduces the sensitivity of Or47b ORNs, which we found to be responsive to 2MC, leading to an increased preference for pheromone-rich males after mating (*Kohlmeier et al., 2021*). Moreover, the finding that 2MC and 7-T induce cAMP levels in pC1b, c neurons in virgin females suggests that virgin females may also use 2MC and 7-T as odorant cues to assess male quality during their first mating. Indeed, females seem to evaluate male quality by the amount of 7-T, as increased 7-T promotes mating receptivity and shortens mating latency (*Grillet et al., 2006*).

Physiological factors like the nutritional status of females prior to mating and the nutritional status of their mates have been shown to influence EHP (*Kim et al., 2023*), and therefore potentially MIES. Hence, it is highly probable that MIES is regulated by additional central neurons such as Dh44-PI neurons that regulate these processes (*Lee et al., 2015*). However, it remains unclear whether and how Dh44-PI neurons and pC1 neurons interact to modulate EHP and MIES. The observation that double knockdown of Dh44R1 and Dh44R2 has only a marginal effect on MIES suggests that Dh44-PI neurons may also function independently of pC1 neurons, raising the possibility that multiple independent central circuits may contribute to the production of MIES.

Our initial screening of ORNs responsible for MIES revealed the involvement of Or47b ORNs, as well as several other ORNs. In addition to 2MC, which acts through Or47b-expressing ORNs, our findings indicate that 7-T and *ppk23* neurons, which are capable of detecting 7-T, also play a role in MIES induction. In *D. melanogaster* and other related species, food odors typically serve as volatile long-range signals that attract both males and females (*Lin et al., 2015*; *Verschut et al., 2023*), suggesting that specific food odors may also influence EHP (*Duménil et al., 2016*). The involvement of multiple ORNs in the regulation of EHP predicts that pC1 neurons may process multiple odorants, not limited to those associated with mating behavior, including food odors. Future studies will explore the full spectrum of odorants processed by pC1 neurons in the regulation of EHP.

In conclusion, we have identified a circuit that, via the detection of a novel male pheromone, potentially signals male quality and governs the female's decision to remove the mating plug of her last mate and mate again.

## Materials and methods

**Key resources table**

| Reagent type (species) or resource | Designation | Source or reference | Identifiers | Additional information |
|---|---|---|---|---|
| Genetic reagent (*D. melanogaster*) | *Canton S* | BDSC | RRID:BDSC_64349 | |
| Genetic reagent (*D. melanogaster*) | $w^{1118}$ | VDRC | VDRC #60000 | |
| Genetic reagent (*D. melanogaster*) | *R71G01 (pC1-Gal4)* | BDSC | RRID:BDSC_39599 | |
| Genetic reagent (*D. melanogaster*) | *Orco1* | BDSC | RRID:BDSC_23129 | |
| Genetic reagent (*D. melanogaster*) | *Or13a-Gal4* | BDSC | RRID:BDSC_9946 | |
| Genetic reagent (*D. melanogaster*) | *Or19a-Gal4* | BDSC | RRID:BDSC_9948 | |
| Genetic reagent (*D. melanogaster*) | *Or23a-Gal4* | BDSC | RRID:BDSC_9955 | |
| Genetic reagent (*D. melanogaster*) | *Or43a-Gal4* | BDSC | RRID:BDSC_9974 | |
| Genetic reagent (*D. melanogaster*) | *Or47b-Gal4* | BDSC | RRID:BDSC_9983 | |
| Genetic reagent (*D. melanogaster*) | *Or47b-Gal4* | BDSC | RRID:BDSC_9984 | |

*Continued on next page*

*Continued*

| Reagent type (species) or resource | Designation | Source or reference | Identifiers | Additional information |
|---|---|---|---|---|
| Genetic reagent (*D. melanogaster*) | *Or65a-Gal4* | BDSC | RRID:BDSC_9993 | |
| Genetic reagent (*D. melanogaster*) | *Or65b-Gal4* | BDSC | RRID:BDSC_23901 | |
| Genetic reagent (*D. melanogaster*) | *Or65c-Gal4* | BDSC | RRID:BDSC_23903 | |
| Genetic reagent (*D. melanogaster*) | *Or67d-Gal4* | BDSC | RRID:BDSC_9998 | |
| Genetic reagent (*D. melanogaster*) | *Or83c-Gal4* | BDSC | RRID:BDSC_23131 | |
| Genetic reagent (*D. melanogaster*) | *Or88a-Gal4* | BDSC | RRID:BDSC_23137 | |
| Genetic reagent (*D. melanogaster*) | *UAS-Or47b* | BDSC | RRID:BDSC_76045 | |
| Genetic reagent (*D. melanogaster*) | *Or47b2/2* | BDSC | RRID:BDSC_51306 | |
| Genetic reagent (*D. melanogaster*) | *Or47b3/3* | BDSC | RRID:BDSC_51307 | |
| Genetic reagent (*D. melanogaster*) | *UAS-TNT active* | BDSC | RRID:BDSC_28837 | |
| Genetic reagent (*D. melanogaster*) | *UAS-TNT inactive* | BDSC | RRID:BDSC_28839 | |
| Genetic reagent (*D. melanogaster*) | *UAS-dTRPA1* | BDSC | RRID:BDSC_26263 | |
| Genetic reagent (*D. melanogaster*) | *UAS-CsChrimson* | BDSC | RRID:BDSC_55135 | |
| Genetic reagent (*D. melanogaster*) | *UAS-GCaMP6m* | BDSC | RRID:BDSC_42748 | |
| Genetic reagent (*D. melanogaster*) | *R52G04-AD* | BDSC | RRID:BDSC_71085 | |
| Genetic reagent (*D. melanogaster*) | *SAG-Gal4 (VT50405)* | VDRC | RRID:Flybase_FBst0489354, VDRC #200652 | |
| Genetic reagent (*D. melanogaster*) | *UAS-Dh44R1-RNAi* | VDRC | RRID:Flybase_FBst0482273, VDRC #110708 | |
| Genetic reagent (*D. melanogaster*) | *UAS-Dh44R2-RNAi* | VDRC | RRID:Flybase_FBst0465025, VDRC #43314 | |
| Genetic reagent (*D. melanogaster*) | *UAS-Dicer2* | VDRC | VDRC #60007 | |
| Genetic reagent (*D. melanogaster*) | *PromE(800)-Gal4* | **Billeter et al., 2009** | N/A | |
| Genetic reagent (*D. melanogaster*) | *UAS-FLP, CRE-F-luc* | **Tanenhaus et al., 2012** | N/A | |
| Genetic reagent (*D. melanogaster*) | *LexAop-FLP* | **Bussell et al., 2014** | N/A | |
| Genetic reagent (*D. melanogaster*) | *UAS-CsChrimson* | **Klapoetke et al., 2014** | N/A | |
| Genetic reagent (*D. melanogaster*) | *UAS-GtACR1* | **Mohammad et al., 2017** | N/A | |

*Continued on next page*

*Continued*

| Reagent type (species) or resource | Designation | Source or reference | Identifiers | Additional information |
|---|---|---|---|---|
| Genetic reagent (*D. melanogaster*) | *UAS-PhotoAC (PACα)* | **Schröder-Lang et al., 2007** | N/A | |
| Genetic reagent (*D. melanogaster*) | *ppk23-Gal4* | **Thistle et al., 2012** | N/A | |
| Genetic reagent (*D. melanogaster*) | *ppk23-* | **Thistle et al., 2012** | N/A | |
| Genetic reagent (*D. melanogaster*) | *ppk28-* | **Thistle et al., 2012** | N/A | |
| Genetic reagent (*D. melanogaster*) | *ppk29-* | **Thistle et al., 2012** | N/A | |
| Genetic reagent (*D. melanogaster*) | *pC1-A* | **Deutsch et al., 2020** | N/A | |
| Genetic reagent (*D. melanogaster*) | *pC1-S* | **Deutsch et al., 2020** | N/A | |
| Genetic reagent (*D. melanogaster*) | *Dh44-pC1-Gal4* | **Kim et al., 2024** | N/A | |
| Genetic reagent (*D. melanogaster*) | *Orco-Gal4* | **Yu et al., 2018** | N/A | |
| Genetic reagent (*D. melanogaster*) | *UAS-EGFP-Orco* | **Yu et al., 2018** | N/A | |
| Genetic reagent (*D. melanogaster*) | *dsx-DBD* | **Wang et al., 2020** | N/A | |
| Genetic reagent (*D. melanogaster*) | *pC1a-split-Gal4* | This study | N/A | |
| Strain, strain background (*Drosophila simulans*) | *Drosophila simulans* | EHIME-Fly, KYORIN-Fly | N/A | |
| Strain, strain background (*Drosophila sechellia*) | *Drosophila sechellia* | EHIME-Fly, KYORIN-Fly | N/A | |
| Strain, strain background (*Drosophila erecta*) | *Drosophila erecta* | EHIME-Fly, KYORIN-Fly | N/A | |
| Strain, strain background (*Drosophila yakuba*) | *Drosophila yakuba* | EHIME-Fly, KYORIN-Fly | N/A | |
| Antibody | Mouse monoclonal anti-Bruchpilot | DSHB | Cat# Nc82; RRID:AB_2314866 | 1:50 |
| Antibody | Rabbit Polyclonal Anti-Green Fluorescent Protein (GFP) | Thermo Fisher Scientific (Invitrogen) | Cat# A-11122, RRID:AB_221569 | 1:1000 |
| Antibody | Alexa 488-conjugated goat anti-rabbit | Thermo Fisher Scientific (Invitrogen) | Cat# A-11008, RRID:AB_143165 | 1:1000 |
| Antibody | Alexa 568-conjugated goat anti-mouse | Thermo Fisher Scientific (Invitrogen) | Cat# A-11004, RRID:AB_2534072 | 1:1000 |
| Chemical compound | Photo-curable UV glue | ThreeBond | A16A01 | |
| Chemical compound | All *trans*-retinal | Sigma-Aldrich | Cat# R2500 | |
| Chemical compound | Vectashield | Vector Laboratories | Cat# H-1000 | |
| Chemical compound | Methyl laurate | Sigma-Aldrich | Cat# W271500 | |
| Chemical compound | 7(Z)-Tricosene | Cayman Chemical | Cat# 9000313 | |
| Chemical compound | *trans*-palmitoleic acid | Cayman Chemical | Cat# 9001798 | |
| Chemical compound | 11-*cis*-vaccenyl acetate (cVA) | Cayman Chemical | Cat# 10010101 | |

*Continued on next page*

*Continued*

| Reagent type (species) or resource | Designation | Source or reference | Identifiers | Additional information |
|---|---|---|---|---|
| Chemical compound | 7(Z)-Pentacosene | Cayman Chemical | Cat# 9000530 | |
| Chemical compound | Beetle Luciferin, Potassium Salt | Promega | Cat# E1601 | |
| Chemical compound | Triton X-100, laboratory grade | Sigma-Aldrich | Cat# X100 | |
| Chemical compound | 2-Methyltetracosane | KIP | N/A | >98%, purity |
| Software and algorithms | Fiji | https://imagej.net/software/fiji/downloads | RRID:SCR_002285 | |
| Software and algorithms | GraphPad Prism9 | https://www.graphpad.com/scientific-software/prism/ | RRID:SCR_002798 | |
| Software and algorithms | Metamorph software | https://www.moleculardevices.com/products/cellular-imaging-systems/acquisition-and-analysis-software/metamorph-microscopy | RRID:SCR_002368 | |
| Software and algorithms | Neuronbridge | https://neuronbridge.janelia.org/ | N/A | |
| Software and algorithms | Computational Morphometry Toolkit (CMTK) | https://github.com/jefferis/fiji-cmtk-gui; *Jefferis, 2015* | RRID:SCR_002234 Version number: v0.1.1 | |
| Software and algorithms | ColorMIP_Mask _Search plugin | https://github.com/JaneliaSciComp/ColorMIP_Mask_Search; *Otsuna et al., 2020* | Version number: v1.0.1 | |
| Other | Digital camcorder | SONY | HDR-CX405 | Behavior recording device |
| Other | Smart phone | Xiaomi | Redmi Note 10 | Behavior recording device |
| Other | Multi-channel LED lights | NeoPixel | Cat# WS2812 | Light activation device; red light, 620–625 nm, 390–420 mcd; green light, 522–525 nm, 660–720 mcd; blue light, 465–467 nm, 180–200 mcd |
| Other | Electron-multiplying CCD camera | Andor Technology | LucaEM R 604M | Calcium imaging assay device |
| Other | Stimulus Controller | Syntech | Type CS-55 | Pheromone delivery device |
| Other | Microplate luminometer | Berthold Technologies | Centro XS3 LB 960 | Luciferase assay device |

## Fly care

Flies were cultured on a standard medium composed of dextrose, corn meal, and yeast, at room temperature on a 12 hr:12 hr light:dark cycle (*Kim et al., 2023*; *Lee et al., 2015*). Behavioral assays were performed at 25°C, except for the thermogenetic activation experiment with dTRPA1. Virgin males and females were collected immediately after eclosion. Males were aged individually for 4–6 days, while females were aged in groups of 15–20. For EHP and mating assays, females were aged for 3–4 days. Assays were performed at Zeitgeber time 3:00–11:00 and were repeated on at least 3 separate days.

## Fly stocks

The following stocks are from the Bloomington *Drosophila* Stock Center (BDSC), the Vienna *Drosophila* Resource Center (VDRC): *Canton S (CS)* (RRID:BDSC_64349), *w1118* (VDRC #60000),

*R71G01* (pC1-Gal4) (RRID:BDSC_39599), *Orco1* (RRID:BDSC_23129), *Or13a-Gal4* (RRID:BDSC_9946), *Or19a-Gal4* (RRID:BDSC_9948), *Or23a-Gal4* (RRID:BDSC_9955), *Or43a-Gal4* (RRID:BDSC_9974), *Or47b-Gal4* (RRID:BDSC_9983), *Or47b-Gal4* (RRID:BDSC_9984), *Or65a-Gal4* (RRID:BDSC_9993), *Or65b-Gal4* (RRID:BDSC_23901), *Or65c-Gal4* (RRID:BDSC_23903), *Or67d-Gal4* (RRID:BDSC_9998), *Or83c-Gal4* (RRID:BDSC_23131), *Or88a-Gal4* (RRID:BDSC_23137), *UAS-Or47b* (RRID:BDSC_76045), *Or47b2/2* (RRID:BDSC_51306), *Or47b3/3* (RRID:BDSC_51307), *UAS-TNT active* (RRID:BDSC_28837), *UAS-TNT inactive* (RRID:BDSC_28839), *UAS-dTRPA1* (RRID:BDSC_26263), *UAS-CsChrimson* (RRID:BDSC_55135), *UAS-GCaMP6m* (RRID:BDSC_42748), *R52G04-AD* (RRID:BDSC_71085), *SAG-Gal4* (VT50405) (RRID:Flybase_FBst0489354, VDRC #200652), *UAS-Dh44R1-RNAi* (RRID:Flybase_FBst0482273, VDRC #110708), *UAS-Dh44R2-RNAi* (RRID:Flybase_FBst0465025, VDRC #43314), *UAS-Dicer2* (VDRC #60007). The following stocks were previously reported: *PromE(800)-Gal4* (*Billeter et al., 2009*), *UAS-FLP, CRE-F-luc* (*Tanenhaus et al., 2012*), *LexAop-FLP* (*Bussell et al., 2014*), *UAS-CsChrimson* (*Klapoetke et al., 2014*), *UAS-GtACR1* (*Mohammad et al., 2017*), *UAS-PhotoAC* (PACα) (*Schröder-Lang et al., 2007*), pC1-A (*Deutsch et al., 2020*), pC1-S (*Deutsch et al., 2020*), *Dh44-pC1-Gal4* (*Kim et al., 2024*), *ppk23-Gal4, ppk23-, ppk28-, ppk29-* (*Thistle et al., 2012*), and *Orco-Gal4, UAS-EGFP-Orco* (*Yu et al., 2018*). *pC1a-split-Gal4* is generated by combining *R52G04-AD* (RRID:BDSC_71085) and *dsx-DBD* (*Wang et al., 2020*). *Drosophila* species other than *D. melanogaster* are obtained from the EHIME-Fly *Drosophila* Stock Center and the KYORIN-Fly *Drosophila* Species Stock Center. To enhance knock-down efficiency, RNAi experiments were performed using flies carrying *UAS-Dicer2* (VDRC #60007). *Table 1* lists the genotypes and number of animals used in all experiments.

## Chemical information

ATR (Cat# R2500), methyl laurate (Cat# W271500), and Triton X-100 (Cat# X100) were obtained from Sigma-Aldrich (St. Louis, MO, USA). The following chemicals were obtained from the Cayman Chemical (Ann Arbor, MI, USA): 7(Z)-Tricosene (CAS No. 52078-42-9, Cat# 9000313), 7(Z)-Pentacosene (CAS No. 63623-49-4, Cat# 9000530), *trans*-palmitoleic acid (CAS No. 10030-73-6, Cat# 9001798), cVA dissolved in EtOH (CAS No. 6186-98-7, Cat# 10010101). 2MC (>98% purity) was custom-synthesized by KIP (Daejeon, Korea). Ethanol is used as a vehicle for 7-T, cVA, *trans*-palmitoleic acid, and methyl laurate, while hexane is used as a vehicle for 2MC.

## Behavior assays

For mating behavior assays, we followed the procedures described previously (*Yapici et al., 2008*). Individual virgin females and naive *CS* males were paired in 10 mm diameter chambers and were recorded using a digital camcorder (SONY, HDR-CX405 or Xiaomi, Redmi Note 10) for either 30 min or 1 hr for the mating assay and 6 hr for the re-mating assay. In the re-mating assay, females that completed their initial mating within 30 min were subsequently paired with naive *CS* males.

To measure EHP, defined as the time elapsed between the end of copulation and sperm ejection, we used the following procedure: Virgin females were individually mated with *CS* males in 10 mm diameter chambers. Following copulation, females were transferred to new chambers, either with or without a *CS* male or pheromone presentation, and their behavior was recorded using a digital camcorder (SONY, HDR-CX405). Typically, females that completed copulation within 30 min were used for analysis. The sperm ejection scene, in which the female expels a white sac containing sperm and the mating plug through the vulva, was directly observed by eye in the recorded video footage. For pheromone presentation, females were individually housed in 10 mm diameter chambers containing a piece of Whatman filter paper (2 mm × 2 mm) treated with 0.5 µl of the pheromone solution and air-dried for 1 min. For thermogenetic activation experiments, females were incubated at the indicated temperatures immediately after the end of copulation. For light activation experiments, a custom-made light activation setup was used with a ring of 104 multi-channel LED lights (NeoPixel, Cat# WS2812; red light, 620–625 nm, 390–420 mcd; green light, 522–525 nm, 660–720 mcd; blue light, 465–467 nm, 180–200 mcd). Females were individually placed in 10 mm diameter chambers, and the chamber was illuminated with light at an intensity of 1100 lux across the chamber during the assay, as measured by an HS1010 light meter. Flies used in these experiments were prepared by culturing them immediately after eclosion in food containing vehicle (EtOH) or 1 mM ATR. They were kept in complete darkness for 3–4 days until the assay was conducted. To prevent the accumulation

**Table 1.** Table of *D. melanogaster* genotypes or *Drosophila* species used to generate the figures and figure supplements in this paper.

| Figure | *D. melanogaster* genotypes or *Drosophila* species of | | | N numbers |
| | Tested females | Mating partner | Incubation partner | From left to right |
|---|---|---|---|---|
| **Figure 1** | | | | |
| **Figure 1B** | w[1118] | Canton-S | Canton-S | 59, 69 |
| **Figure 1C** | w[1118] | Canton-S | Canton-S | 18, 15 |
| **Figure 1D** | w[1118] | Canton-S | Canton-S | 12, 12 |
| **Figure 1E** | w[1118] | Canton-S | Canton-S | 20, 18, 23 |
| **Figure 1F** | w[1118]; TI{w[+mW.hs]=TI}Orco[1] | Canton-S | Canton-S | 55, 47 |
| **Figure 2** | | | | |
| **Figure 2A** | w[1118];Or47b-Gal4/UAS-TNTinactive(P{UAS-TeTxLC.(-)Q}A2) | Canton-S | Canton-S | 14, 14 |
| | w[1118];Or47b-Gal4/UAS-TNTactive (P{w[+mC]=UAS-TeTxLC.tnt}E2) | Canton-S | Canton-S | 16, 18 |
| **Figure 2B** | w[1118];Or47b-Gal4/+ | Canton-S | Canton-S | 28, 31 |
| | w[1118];;UAS-dTRPA1/+ | Canton-S | Canton-S | 21, 16 |
| | w[1118];Or47b/+;UAS-dTRPA1/+ | Canton-S | Canton-S | 11, 15 |
| **Figure 2C** | w[1118]; TI{w[+mW.hs]=TI}Orco [1] | Canton-S | Canton-S | 12, 12 |
| | w[1118]; Or47b-Gal4>UAS-EGFP-Orco; Orco[1]/Orco[1] | Canton-S | Canton-S | 13, 14 |
| **Figure 2D** | w[1118]; Or47b[2]/+ | Canton-S | Canton-S | 12, 15 |
| | w[1118]; Or47b[3]/+ | Canton-S | Canton-S | 13, 14 |
| | w[1118]; Or47b[2]/Or47b[3] | Canton-S | Canton-S | 13, 12 |
| **Figure 2E** | w[1118]; Or47b[2]/Or47b[2] | Canton-S | Canton-S | 14, 15 |
| | w[1118]; Or47b-Gal4>P{w[+mC]=UAS-Or47b.MYC}2; Or47b[2]/Or47b[2] | Canton-S | Canton-S | 11, 11 |
| **Figure 3** | | | | |
| **Figure 3A** | w[1118] | Canton-S | | 13, 16 |
| **Figure 3B** | w[1118]; TI{w[+mW.hs]=TI}Orco[1] | Canton-S | | 11, 12 |
| **Figure 3C** | w[1118]; TI{w[+mW.hs]=TI}Or47b[2] | Canton-S | | 14, 17 |
| **Figure 3D** | w[1118];Or47b-Gal4/+;Orco[1]/Orco[1] | Canton-S | | 22, 22 |
| | w[1118];UAS-Orco/+; Orco[1]/Orco[1] | Canton-S | | 15, 14 |
| | w[1118];Or47b-Gal4/UAS-Orco;Orco[1]/Orco[1] | Canton-S | | 18, 19 |
| **Figure 4** | | | | |
| **Figure 4A** | w[1118] | Canton-S | Canton-S | 18, 22 |
| **Figure 4B** | w[1118] | Canton-S | | 23, 31 |
| **Figure 4C** | w[1118] | Canton-S | | 16, 17 |

*Table 1 continued on next page*

*Table 1 continued*

| Figure | *D. melanogaster* genotypes or *Drosophila* species of | | | N numbers |
|---|---|---|---|---|
| **Figure 4D** | w[1118];ppk23-Gal4/UAS-TNTinactive(P{UAS-TeTxLC.(-)Q}A2) | Canton-S | Canton-S | 18, 13 |
| | w[1118];ppk23-Gal4/UAS-TNTactive (P{w[+mC]=UAS-TeTxLC.tnt}E2) | Canton-S | Canton-S | 17, 17 |
| | | | | |
| **Figure 5** | | | | |
| **Figure 5A** | w[1118];pC1(R71G01)-AD/+;Dsx-DBD/UAS-GtACR1 | Canton-S | | 22, 21 |
| **Figure 5B** | w[1118];VT25602-AD/+;UAS-GtACR1/VT2064-DBD | Canton-S | | 18, 18 |
| **Figure 5C** | w[1118];R52G04-AD/+;UAS-GtACR1/Dsx-DBD | Canton-S | | 15, 14 |
| **Figure 5D** | w[1118];;Dh44-pC1 (Dsx-DBD, Dh44A-AD)-GAL4/UAS-GtACR1 | Canton-S | | 17, 20 |
| **Figure 5E** | w[1118];UAS-FLP/+; GMR71G01-Gal4, CRE-F-Luc/+ | Canton-S | | 12, 12, 12 |
| | w[1118];R52G04-AD/+;UAS-FLP, CRE-F-Luc/Dsx-DBD | Canton-S | | 12, 12, 12 |
| | w[1118];;UAS-FLP, CRE-F-Luc/Dh44-pC1 (Dsx-DBD, Dh44A-AD)-GAL4 | Canton-S | | 16, 16, 16 |
| | w[1118];VT25602-AD/+;UAS-FLP, CRE-F-Luc/VT2064-DBD | Canton-S | | 12, 12, 12 |
| **Figure 5F** | w[1118]; Or47b[2]/+; pC1-FLP, CRE-F-Luc | Canton-S | | 8, 8, 8 |
| | w[1118]; Or47b[2]/Or47b[2]; pC1-FLP, CRE-F-Luc | Canton-S | | 8, 8, 8 |
| | w[1118]; Or47b[3]/+; pC1-FLP, CRE-F-Luc | Canton-S | | 8, 8, 8 |
| | w[1118]; Or47b[3]/Or47b[3]; pC1-FLP, CRE-F-Luc | Canton-S | | 8, 8, 8 |
| | | | | |
| **Figure 6A** | w[1118]; R52G04-AD/+;UAS-PhotoAC/Dsx-DBD | Canton-S | | 18, 22 |
| | w[1118];;UAS-PhotoAC/Dh44-pC1 (Dsx-DBD, Dh44A-AD)-GAL4 | Canton-S | | 22, 28 |
| | w[1118]; VT25602-AD/+;UAS-PhotoAC/VT2064-DBD | Canton-S | | 21, 20 |
| **Figure 6B** | w[1118];UAS-GCaMP6m/+; pC1(GMR71G01)-GAL4/UAS-PhotoAC | Canton-S | | 9, 9, 9 |
| **Figure 6C** | w[1118];;+/UAS-PhotoAC | Canton-S (1st, 2nd) | | 60 |
| | w[1118];;Dh44-pC1 (Dsx-DBD, Dh44A-AD)-GAL4/UAS-PhotoAC | Canton-S (1st, 2nd) | | 18 |
| | | | | |
| **Figure 2—figure supplement 1** | w[1118];+/P{w[+mC]=UAS-TeTxLC.tnt}E2 | Canton-S | Canton-S | 27, 27 |
| | w[1118];Or13a-Gal4/P{w[+mC]=UAS-TeTxLC.tnt}E2 | Canton-S | Canton-S | 9, 12 |
| | w[1118];+/P{w[+mC]=UAS-TeTxLC.tnt}E2;+/Or19a-Gal4 | Canton-S | Canton-S | 8, 6 |
| | w[1118];+/P{w[+mC]=UAS-TeTxLC.tnt}E2;+/Or23a-Gal4 | Canton-S | Canton-S | 11, 11 |
| | w[1118];+/P{w[+mC]=UAS-TeTxLC.tnt}E2;+/Or43a-Gal4 | Canton-S | Canton-S | 18, 16 |
| | w[1118];+/P{w[+mC]=UAS-TeTxLC.tnt}E2;+/Or47b-Gal4 | Canton-S | Canton-S | 12, 11 |
| | w[1118];Or65a-Gal4/P{w[+mC]=UAS-TeTxLC.tnt}E2 | Canton-S | Canton-S | 13, 16 |
| | w[1118];Or65b-Gal4/P{w[+mC]=UAS-TeTxLC.tnt}E2 | Canton-S | Canton-S | 18, 14 |
| | w[1118];+/P{w[+mC]=UAS-TeTxLC.tnt}E2;+/Or65c-Gal4 | Canton-S | Canton-S | 20, 18 |
| | w[1118];Or67d-Gal4/P{w[+mC]=UAS-TeTxLC.tnt}E2 | Canton-S | Canton-S | 15, 19 |

*Table 1 continued on next page*

*Table 1 continued*

| Figure | *D. melanogaster* genotypes or *Drosophila* species of | | | N numbers |
|---|---|---|---|---|
| | *w[1118];Or83c-Gal4/P{w[+mC]=UAS-TeTxLC.tnt}E2* | *Canton-S* | *Canton-S* | 20, 17 |
| | *w[1118];Or88a-Gal4/P{w[+mC]=UAS-TeTxLC.tnt}E2* | *Canton-S* | *Canton-S* | 21, 19 |
| *Figure 3—figure supplement 1* | | | | |
| *Figure 3—figure supplement 1A* | *w[1118]* | *Canton-S* | | 26, 14, 18, 13 |
| *Figure 3—figure supplement 1B* | *w[1118]* | *Canton-S* | | 22, 14, 12, 12 |
| *Figure 3—figure supplement 2* | | | | |
| *Figure 3—figure supplement 2A* | *w[1118]* | *Canton-S* | | 33 |
| | *w[1118]* | *Canton-S* | *+;PromE(800)-Gal4/UAS-Tra* | 36 |
| | *w[1118]* | *Canton-S* | *+;PromE(800)-Gal4/UAS-Tra-RNAi* | 17 |
| *Figure 3—figure supplement 2B* | *w[1118]* | *Canton-S* | | 20 |
| | *w[1118]* | *Canton-S* | *D. melanogaster* | 23 |
| | *w[1118]* | *Canton-S* | *D. simulans* | 21 |
| | *w[1118]* | *Canton-S* | *D. sechellia* | 20 |
| | *w[1118]* | *Canton-S* | *D. erecta* | 19 |
| | *w[1118]* | *Canton-S* | *D. yakuba* | 21 |
| *Figure 3—figure supplement 3* | *w[1118]* | *Canton-S* | | 18, 18, 14, 17, 15, 13 |
| *Figure 4—figure supplement 1* | | | | |
| *Figure 4—figure supplement 1A* | *w[1118]* | *Canton-S* | | 17, 17, 18, 18, 17, 18 |
| *Figure 4—figure supplement 1B* | *w[1118]* | *Canton-S* | | 15, 15, 15, 16, 14, 15 |
| *Figure 4—figure supplement 1C* | *w[1118]* | *Canton-S* | *Canton-S* | 21, 18 |
| | *w[1118]/ppk23-* | *Canton-S* | *Canton-S* | 17, 21 |
| | *ppk23-* | *Canton-S* | *Canton-S* | 13, 15 |
| *Figure 4—figure supplement 1D* | *w[1118]* | *Canton-S* | *Canton-S* | 18, 14 |
| | *w[1118]/ppk28-* | *Canton-S* | *Canton-S* | 23, 25 |
| | *ppk28-* | *Canton-S* | *Canton-S* | 22, 17 |
| *Figure 4—figure supplement 1E* | *w[1118]* | *Canton-S* | *Canton-S* | 17, 14 |
| | *w[1118];ppk29-/+* | *Canton-S* | *Canton-S* | 19, 20 |

*Table 1 continued on next page*

Table 1 continued

| Figure | *D. melanogaster* genotypes or *Drosophila* species of | | | N numbers |
|---|---|---|---|---|
| | *ppk29-* | Canton-S | Canton-S | 16, 17 |
| Figure 5—figure supplement 1A–C | w[1118];R52G04-AD/UAS-myrGFP;Dsx-DBD/UAS-myrGFP | | | |
| Figure 5—figure supplement 1D | w[1118];R52G04-AD/+;Dsx-DBD/UAS-GtACR1 | Canton-S | | 82, 60 |
| Figure 5—figure supplement 2A | w[1118];UAS-FLP/+; GMR71G01-Gal4, CRE-F-Luc/+ | | | 8, 8, 8, 12, 8, 4 |
| Figure 5—figure supplement 2B | w[1118];UAS-FLP/+; GMR71G01-Gal4, CRE-F-Luc/+ | | | 8, 8, 8, 12, 8, 4 |
| Figure 5—figure supplement 3 | w[1118];UAS-FLP/+; GMR71G01-Gal4, CRE-F-Luc/+ | | | 12, 12, 12 |
| | w[1118]; R52G04-AD/+;UAS-FLP, CRE-F-Luc/Dsx-DBD | | | 12, 12, 12 |
| | w[1118];;UAS-FLP, CRE-F-Luc/Dh44-pC1 (Dsx-DBD, Dh44A-AD)-GAL4 | | | 12, 12, 12 |
| | w[1118]; VT25602-AD/+;UAS-FLP, CRE-F-Luc/VT2064-DBD | | | 10, 10, 10 |
| Figure 6—figure supplement 1 | w[1118]/UAS-Dcr2;;GMR71G01-Gal4/+ | Canton-S | Canton-S | 27, 26 |
| | w[1118];UAS-Dh44R1-RNAi/+; UAS-Dh44R2-RNAi/+ | Canton-S | Canton-S | 18, 17 |
| | w[1118]/UAS-Dicer2;UAS-Dh44R1-RNAi1/+; GMR71G01-GAL4/UAS-Dh44R2-RNAi2 | Canton-S | Canton-S | 35, 30 |

of residual pheromones, all behavioral chambers were cleaned with 70% water/ethanol or acetone before and after the experiment.

## Calcium imaging

We followed the procedures described previously (*Kim et al., 2024*; *Kohatsu and Yamamoto, 2015*). Following copulation, freshly mated female flies were temporarily immobilized using ice anesthesia, and their heads were attached to a custom-made thin metal plate with a 1 mm diameter hole using photo-curable UV glue (ThreeBond, A16A01). An opening in the fly's head was created using a syringe needle under saline (108 mM NaCl, 5 mM KCl, 2 mM $CaCl_2$, 8.2 mM $MgCl_2$, 4 mM $NaHCO_3$, 1 mM $NaH_2PO_4$, 5 mM trehalose, 10 mM sucrose, 5 mM HEPES pH 7.5). Imaging was performed with a Zeiss Axio Examiner A1 microscope equipped with an electron-multiplying CCD camera (Andor Technology, Luca[EM] R 604M) and an LED light source (CoolLED, Precis Excite). Metamorph software (Molecular Devices, RRID:SCR_002368) was used for image analysis. The Syntech Stimulus Controller (Type CS-55) was used to deliver the male pheromone using an airflow. 2 μl of pheromone solution was applied to a piece of Whatman filter paper (2 mm × 1 mm), which was then inserted into a glass Pasteur pipette after solvent evaporation.

## Luciferase assay

We followed the procedures described previously (*Kim et al., 2024*; *Tanenhaus et al., 2012*). For the assay, 3-day-old virgin females or freshly mated females were used. A group of three fly heads, kept at –80°C, was homogenized using cold homogenization buffer (15 mM HEPES, 10 mM KCl, 5 mM $MgCl_2$, 0.1 mM EDTA, 0.5 mM EGTA). Luciferase activity was measured using beetle luciferin potassium salt (Promega, Cat# E1603) and a microplate luminometer (Berthold Technologies, Centro $XS^3$ LB 960), following the manufacturer's instructions. For pheromone presentation, flies were placed in 10 mm

diameter chambers containing a piece of Whatman filter paper (4 mm × 6 mm) treated with 1 µl of the pheromone solution and air-dried for 1 min.

## Immunohistochemistry

3- to 5-day-old virgin female flies were dissected in phosphate-buffered saline (PBS) and fixed in 4% paraformaldehyde in PBS for 30 min at room temperature. After fixation, the brains were thoroughly washed in PBST (0.1% Triton X-100 in PBS) and then blocked with 5% normal goat serum in PBST. After blocking, brains were incubated with primary antibody in PBST for 48 hr at 4°C, washed with PBST, and then incubated with secondary antibody in PBST for 24 hr at 4°C. The samples were washed three times with PBST and once with PBS before mounting in Vectashield (Vector Laboratories, Cat# H-1000). Antibodies used were rabbit anti-GFP (1:1000; Thermo Fisher Scientific, Cat# A-11122, RRID:AB_221569), mouse anti-nc82 (1:50; Developmental Studies Hybridoma Bank, Cat# Nc82; RRID:AB_2314866), Alexa 488-conjugated goat anti-rabbit (1:1000; Thermo Fisher Scientific, Cat# A-11008, RRID:AB_143165), Alexa 568-conjugated goat anti-mouse (1:1000; Thermo Fisher Scientific, Cat# A-11004, RRID:AB_2534072). Brain images were acquired with a Zeiss LSM 700/Axiovert 200M (Zeiss) and processed with Fiji (https://imagej.net/software/fiji/downloads, RRID:SCR_002285).

## Color depth MIP-based anatomical analysis

A stack of confocal images of *pC1a-split-Gal4>UAS-myr-EGFP* adult female brains stained with anti-GFP and anti-nc82 was used. Images were registered to the JRC2018 unisex brain template (*Bogovic et al., 2020*) using the Computational Morphometry Toolkit (CMTK, https://github.com/jefferis/fiji-cmtk-gui RRID:SCR_002234, v0.1.1). Color depth MIP masks of *pC1a-split-Gal4* neurons and pC1a (ID, 5813046951) in Hemibrain (*Scheffer et al., 2020*; *Figure 5—figure supplement 1C*) were generated using the ColorMIP_Mask_Search plugin for Fiji (*Otsuna et al., 2018*; https://github.com/JaneliaSciComp/ColorMIP_Mask_Search) and NeuronBridge (*Clements et al., 2022*; https://neuronbridge.janelia.org/). Similarity score and rank were calculated using NeuronBridge.

## Statistical analysis

Statistical analysis was conducted using GraphPad Prism 9 (GraphPad, RRID:SCR_002798), with specific details of each statistical method provided in the figure legends.

## Acknowledgements

We thank S Kang, J-H Yoon, and B Lee for excellent technical assistance and the GIST Advanced Institute of Instrumental Analysis (GAIA) for the confocal microscopy analysis. Fly stocks were obtained from the Bloomington *Drosophila* Stock Center (NIH P40OD018537), the Vienna *Drosophila* Resource Center (VDRC), the Kyoto Stock Center, the EHIME-Fly *Drosophila* Species Stock Center, the KYORIN-Fly *Drosophila* Species Stock Center, and the Korea *Drosophila* Resource Center (NRF-2022M3H9A1085169). This work was supported by National Research Foundation of Korea grants NRF-2022R1A2C3008091 (Y-JK), NRF-2022M3E5E8081194 (Y-JK), NRF-2019R1A4A1029724 (Y-JK), NRF-2017R1A6A3A11027866 (D-HK), NRF-2021R1I1A1A01060304 (D-HK), GIST Research Institute (GRI) GIST-MIT research collaboration grant funded by GIST in 2023 (Y-JK), 2022, 2023 AI-based GIST Research Scientist Project (D-HK).

## Additional information

### Funding

| Funder | Grant reference number | Author |
|---|---|---|
| National Research Foundation of Korea | NRF-2022M3H9A1085169 | Young-Joon Kim |
| National Research Foundation of Korea | NRF-2022R1A2C3008091 | Young-Joon Kim |

| Funder | Grant reference number | Author |
| --- | --- | --- |
| National Research Foundation of Korea | NRF-2022M3E5E8081194 | Young-Joon Kim |
| National Research Foundation of Korea | NRF-2019R1A4A1029724 | Young-Joon Kim |
| National Research Foundation of Korea | NRF-2017R1A6A3A11027866 | Do-Hyoung Kim |
| National Research Foundation of Korea | NRF-2021R1I1A1A01060304 | Do-Hyoung Kim |
| Gwangju Institute of Science and Technology | GIST Research Institute (GRI) GIST-MIT research collaboration grant | Young-Joon Kim |
| Gwangju Institute of Science and Technology | AI-based GIST Research Scientist Project | Do-Hyoung Kim |

The funders had no role in study design, data collection and interpretation, or the decision to submit the work for publication.

## Author contributions

Minsik Yun, Conceptualization, Investigation, Writing – original draft; Do-Hyoung Kim, Tal Soo Ha, Eungyu Park, Investigation; Kang-Min Lee, Methodology; Markus Knaden, Bill S Hansson, Methodology, Writing – review and editing; Young-Joon Kim, Conceptualization, Supervision, Funding acquisition, Writing – original draft, Writing – review and editing

## Author ORCIDs

Minsik Yun http://orcid.org/0000-0002-0011-0942
Tal Soo Ha https://orcid.org/0000-0003-0015-6075
Markus Knaden https://orcid.org/0000-0002-6710-1071
Bill S Hansson https://orcid.org/0000-0002-4811-1223
Young-Joon Kim https://orcid.org/0000-0002-7990-754X

Reviewer #1 (Public Review): https://doi.org/10.7554/eLife.96013.3.sa1
Reviewer #2 (Public Review): https://doi.org/10.7554/eLife.96013.3.sa2
Author response https://doi.org/10.7554/eLife.96013.3.sa3

# Additional files

## Supplementary files

- MDAR checklist
- Source data 1. Source data for all figures and figure supplements.

## Data availability

All data generated or analysed during this study are included in the manuscript and supporting files.

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
