## [Editor Report · eLife assessment]

This **important** work unravels how female *Drosophila* can assess their social context via chemosensory cues and modulate the sperm storage process after copulation accordingly. A **compelling** set of rigorous experiments uncovers specific pheromones that influence the excitability of the female brain receptivity circuit and their propensity to discard inseminate from a mating. This insight into neuronal mechanisms of sexual behavior plasticity is of general interest to scientists working in the fields of animal behavior, neuroscience, evolution, and sexual selection, as well as insect chemosensation and reproduction.

---

## [Referee Report · Reviewer #1 (Public Review)]

Yun et al. examined the molecular and neuronal underpinnings of changes in *Drosophila* female reproductive behaviors in response to social cues. Specifically, the authors measure the ejaculate-holding period, which is the amount of time females retain male ejaculate after mating (typically 90 min in flies). They find that female fruit flies, *Drosophila melanogaster*, display shorter holding periods in the presence of a native male or male-associated cues, including 2-Methyltetracosane (2MC) and 7-Tricosene (7-T). They further show that 2MC functions through Or47b olfactory receptor neurons (ORNs) and the Or47b channel, while 7-T functions through ppk23 expressing neurons. Interestingly, their data also indicates that two other olfactory ligands for Or47b (methyl laurate and palmitoleic acid) do not have the same effects on the ejaculate-holding period. By performing a series of behavioral and imaging experiments, the authors reveal that an increase in cAMP activity in pC1 neurons is required for this shortening of the ejaculate-holding period and may be involved in the likelihood of remating. This work lays the foundation for future studies on sexual plasticity in female Drosophila.

The conclusions of this paper are supported by the data and the authors have revised the manuscript in accordance with comments of the reviewers. This revised version also contains the expression pattern of the lines used for modulating individual pC1 subtypes. These data and reagents open interesting avenues for future studies on female receptivity and mate choice.

---

## [Referee Report · Reviewer #2 (Public Review)]

The work by Yun et al. explores an important question related to post-copulatory sexual selection and sperm competition: Can females actively influence the outcome of insemination by a particular male by modulating storage and ejection of transferred sperm in response to contextual sensory stimuli? The present work is exemplary for how the *Drosophila* model can give detailed insight in basic mechanism of sexual plasticity, addressing the underlying neuronal circuits on a genetic, molecular and cellular level.

Using the *Drosophila* model, the authors show that the presence of other males or mated females after mating shortens the ejaculate-holding period (EHP) of a female, i.e. the time she takes until she ejects the mating plug and unstored sperm. Through a series of thorough and systematic experiments involving the manipulation of olfactory and chemogustatory neurons and genes in combination with exposure to defined pheromones, they uncover two pheromones and their sensory cells for this behavior. Exposure to the male specific pheromone 2MC shortens EHP via female Or47b olfactory neurons, and the contact pheromone 7-T, present males and on mated females, does so via ppk23 expressing gustatory foreleg neurons. Both compounds increase cAMP levels in a specific subset of central brain receptivity circuit neurons, the pC1b,c neurons. By employing an optogenetically controlled adenyl cyclase, the authors show that increased cAMP levels in pC1b,c neurons increase their excitability upon male pheromone exposure, decrease female EHP and increase the remating rate. This provides convincing evidence for the role of pC1b,c neurons in integrating information about the social environment and mediating not only virgin, but also mated female post-copulatory mate choice.

Understanding context and state-dependent sexual behavior is of fundamental interest. Mate behavior is highly context-dependent. In animals subjected to sperm competition, the complexities of optimal mate choice have attracted a long history of sophisticated modelling in the framework of game theory. These models are in stark contrast to how little we understand so far about the biological and neurophysiological mechanisms of how females implement post-copulatory or so-called "cryptic" mate choice and bias sperm usage when mating multiple times.

The strength of the paper is decrypting "cryptic" mate choice, i.e. the clear identification of physiological mechanisms and proximal causes for female post-copulatory mate choice. The discovery of peripheral chemosensory nodes and of neurophysiological mechanisms in central circuit nodes will provide a fruitful starting point to fully map the circuits for female receptivity and mate choice during the whole gamut of female life history.

---

## [Author Response]

The following is the authors’ response to the original reviews.

**Public Reviews:**

**Reviewer #1 (Public Review):**
Yun et al. examined the molecular and neuronal underpinnings of changes in *Drosophila* female reproductive behaviors in response to social cues. Specifically, the authors measure the ejaculate-holding period, which is the amount of time females retain male ejaculate after mating (typically 90 min in flies). They find that female fruit flies, *Drosophila melanogaster*, display shorter holding periods in the presence of a native male or male-associated cues, including 2-Methyltetracosane (2MC) and 7-Tricosene (7-T). They further show that 2MC functions through Or47b olfactory receptor neurons (ORNs) and the Or47b channel, while 7-T functions through ppk23 expressing neurons. Interestingly, their data also indicates that two other olfactory ligands for Or47b (methyl laurate and palmitoleic acid) do not have the same effects on the ejaculate-holding period. By performing a series of behavioral and imaging experiments, the authors reveal that an increase in cAMP activity in pC1 neurons is required for this shortening of the ejaculate-holding period and may be involved in the likelihood of remating. This work lays the foundation for future studies on sexual plasticity in female *Drosophila*.The conclusions of this paper are mostly supported by the data, but aspects of the lines used for individual pC1 subtypes and visual contributions as well as the statistical analysis need to be clarified.(1) The pC1 subtypes (a - e) are delineated based on their morphology and connectivity. While the morphology of these neurons is distinct, they do share a resemblance that can be difficult to discern depending on the imaging performed. Additionally, genetic lines attempting to label individual neurons can easily be contaminated by low-level expression in off-target neurons in the brain or ventral nerve cord (VNC), which could contribute to behavioral changes following optogenetic manipulations. In Figures 5C - D, the authors generated and used new lines for labeling pC1a and pC1b+c. The line for pC1b+c was imaged as part of another recent study (https://doi.org/10.1073/pnas.2310841121). However, similar additional images of the pC1a line (i.e. 40x magnification and VNC expression) would be helpful in order to validate its specificity.

We have included the high-resolution images of the expression of the pC1a-split-Gal4 driver in the brain and the VNC in the new figures S6A and S6B.

(2) The author's experiments examining olfactory and gustatory contributions to the holding period were well controlled and described. However, the experiments in Figure 1D examining visual contributions were not sufficiently convincing as the line used (w1118) has previously been shown to be visually impaired (Wehner et al., 1969; Kalmus 1948). Using another wild-type line would have improved the authors' claims.

It is evident that w1118 flies are visually impaired and are able to receive a limited amount of visual information in dim red light. Nevertheless, they are able to exhibit MIES phenotypes, which further supports the dispensability of visual information in MIES. In a 2024 study, Doubovetzky et al. (1) found that MIES in ninaB mutant females, which have defects in visual sensation, was not altered. This further corroborates our assertion that vision is likely to be of lesser importance than olfaction in MIES.

(3) When comparisons between more than 2 groups are shown as in Figures 1E, 3D, and 5E, the comparisons being made were not clear. Adding in the results of a nonparametric multiple comparisons test would help for the interpretation of these results.

We have revised figures 1E, 3D, 5E and the accompanying legends as suggested.

**Reviewer #2 (Public Review):**
The work by Yun et al. explores an important question related to post-copulatory sexual selection and sperm competition: Can females actively influence the outcome of insemination by a particular male by modulating the storage and ejection of transferred sperm in response to contextual sensory stimuli? The present work is exemplary for how the *Drosophila* model can give detailed insight into the basic mechanism of sexual plasticity, addressing the underlying neuronal circuits on a genetic, molecular, and cellular level.Using the *Drosophila* model, the authors show that the presence of other males or mated females after mating shortens the ejaculate-holding period (EHP) of a female, i.e. the time she takes until she ejects the mating plug and unstored sperm. Through a series of thorough and systematic experiments involving the manipulation of olfactory and chemo-gustatory neurons and genes in combination with exposure to defined pheromones, they uncover two pheromones and their sensory cells for this behavior. Exposure to the male-specific pheromone 2MC shortens EHP via female Or47b olfactory neurons, and the contact pheromone 7-T, present in males and on mated females, does so via ppk23 expressing gustatory foreleg neurons. Both compounds increase cAMP levels in a specific subset of central brain receptivity circuit neurons, the pC1b,c neurons. By employing an optogenetically controlled adenyl cyclase, the authors show that increased cAMP levels in pC1b and c neurons increase their excitability upon male pheromone exposure, decrease female EHP, and increase the remating rate. This provides convincing evidence for the role of pC1b,c neurons in integrating information about the social environment and mediating not only virgin but also mated female post-copulatory mate choice.Understanding context and state-dependent sexual behavior is of fundamental interest. Mate behavior is highly context-dependent. In animals subjected to sperm competition, the complexities of optimal mate choice have attracted a long history of sophisticated modelling in the framework of game theory. These models are in stark contrast to how little we understand so far about the biological and neurophysiological mechanisms of how females implement post-copulatory or so-called "cryptic" mate choice and bias sperm usage when mating multiple times.The strength of the paper is decrypting "cryptic" mate choice, i.e. the clear identification of physiological mechanisms and proximal causes for female post-copulatory mate choice. The discovery of peripheral chemosensory nodes and neurophysiological mechanisms in central circuit nodes will provide a fruitful starting point to fully map the circuits for female receptivity and mate choice during the whole gamut of female life history.

We appreciate the positive response to our work.

Recommendations for the authors:
**Reviewing Editor (Recommendations For The Authors):**
While appreciating the quality of the work the reviewers had a few key concerns that would greatly improve the manuscript. These are:(1) In some cases the specific statistical analyses are not clear. Could the authors please clarify what comparisons were made and the specific tests used?

We have clarified the comparisons made in the multiple comparison analysis and specified the tests used in figures 1E, 3D, 5E.

(2) Could the authors please include data that verify the expression patterns of their new reagent for pC1a, which will be useful for the community?

Figure S6 was revised to include the expression of the pC1a-split-Gal4 gene in the brain (Fig. S6A) and the VNC (Fig. S6B).

(3) A figure summarising their findings in the context of known circuitry will be useful.

A new Figure 7 has been prepared, which provides a summary of our findings.

(4) The SAG data are interesting. Do the authors wish to consider moving it to the main text or removing it if too preliminary?

The supplementary figure 10 and related discussions in the discussion section have been removed.

In the revised version of this manuscript, we present new evidence that the Or47b gene is required for 2MC-induced cAMP elevation in pC1 neurons, but not for 7T-induced one (see Fig. 5F). This observation supports that Or47b is a receptor for 2MC.

The following paragraph was inserted at line 248 to provide a detailed description of the new findings: "To further test the role of Or47b in 2MC detection, we generated Or47b-deficient females with pC1 neurons expressing the CRE-luciferase reporter. Females with one copy of the wild-type Or47b allele, which served as the control group, showed robust CRE-luciferase reporter activity in response to either 2MC or 7-T. In contrast, *Or47b*-deficient females showed robust CRE-luciferase activity in response to to 7-T, but little activity in response to 2MC. This observation suggests that the odorant receptor Or47b plays an essential role in the selective detection of 2MC (Fig. 5F).”

In addition, the following sentence was inserted at line 308 in the discussion section: “In this study, we provide compelling evidence that 2MC induces cAMP elevation in pC1 neurons and EHP shortening via both the Or47b receptor and Or47b ORNs, suggesting that 2MC functions as an odorant ligand for Or47b.”

Relative CRE-luciferase reporter activity of pC1 neurons in females of the indicated genotypes, incubated with a piece of filter paper perfumed with solvent vehicle control or the indicated pheromones immediately after mating. The CRE-luciferase reporter activity of pC1 neurons of Or47b-deficient females (*Or47b2/2* or *Or47b3/3*) was observed to increase in response to 7-T but not to 2MC. To calculate the relative luciferase activity, the average luminescence unit values of the female incubated with the vehicle are set to 100%. Mann-Whitney Test (n.s. *p* > 0.05; ***p* < 0.01; ****p* < 0.001; *****p* < 0.0001). Gray circles indicate the relative luciferase activity (%) of individual females, and the mean ± SEM of data is presented.

**Reviewer #1 (Recommendations For The Authors):**
(1) There was a discrepancy between the text and the figures. Based on the asterisks above the data in Figure S5A, the data supports only 150 ng of 7-T shortening the ejaculation holding period. However, the text states that (line 190) "150 or 375 ng of 7-T significantly shortened EHP." It would be helpful if the authors clarified this discrepancy.

The sentence has been revised and now reads as follows: ‘150 ng of 7-T significantly shortened EHP’.

(2) Based on the current organization of the text, it was not clear how 2MC was identified and its concentrations were known to be physiologically relevant. It would be helpful if the authors could expand on this in lines 178 - 179.

The following sentences were inserted into the revised version of the manuscript at line 178: The EHP was therefore measured in females incubated in a small mating chamber containing a piece of filter paper perfumed with male CHCs, including 2-methylhexacosane, 2-methyldocosane, 5-methyltricosane, 7-methyltricosane, 10Z-heneicosene, 9Z-heneicosene, and 2MC at various concentrations (not shown). Among these, 2MC at 750 ng was the only one that significantly reduced EHP (Fig. 3A; Fig. S4). 2MC was mainly found in males, but not in virgin females (30). Notably, it is present in *D. melanogaster*, *D. simulans*, *D. sechellia*, and *D. erecta*, but not in *D. yakuba* (30, 60).

(3) The inset pie chart image illustrating MIES in Figure 1A was difficult to interpret. It would be helpful if the authors used a different method for representing this (i.e. a timeline).

Figure 1A was revised as suggested.

(4) In lines 121 - 122, the authors state that the females are exposed to "actively courting naive wild type Canton S males." This was difficult to understand and might be improved by removing "actively courting."

Revised as suggested.

**Reviewer #2 (Recommendations For The Authors):**
(1) Summary figureThe story is quite comprehensive and contains a lot of detail regarding the interaction of signaling pathways, internal state, and sensory stimuli. I believe a schematic summary figure bringing together all findings could be very helpful and would make it much easier to understand the discussion!

Figure 7 has been prepared, which provides a summary of the findings and an explanation of the current working model.

(2) Figure S10/effect on SAG activation of EHPAt the moment, the quite interesting and relevant result that SAG activation shortens EHP shown in Figure S10 is only referred to in the discussion. Maybe move this to the results and give it a bit more attention? Actually, I believe this is a very exciting finding that could also be the basis for some more interesting speculations about physiological relevance. Since SAG is silenced upon seminal fluid/sex peptide exposure after mating, a mating with failed SAG silencing (i.e. unusually high post-mating SAG activity) could indicate to the female that there was low or failed sex peptide/seminal fluid transfer. In such a case it would be probably advantageous for the female to decrease EHP and quickly remate, as females need the "beneficial" effects of seminal fluid on ovulation and physiology adaptation. SAG could therefore represent another arm of sensing male quality- here not via external pheromones, but internally, via sensing male sex peptide levels.If this is a bit preliminary and rather suited to start a new study, Figure S10 could also be removed from the current manuscript.

Figure S10 and associated text were removed in the revised version of the manuscript.

(3) PhotoAC experiments in pC1b,c: the authors find that raising cAMP levels in pC1b,c leads to a decrease in EHP. They argue that increased cAMP levels lead to higher excitability of pC1b,c. This implies that the activity of pC1b,c promotes mating plug ejection. I assume the authors have also tried activating pC1b,c directly by optogenetic cation channels? What is the outcome of this? If different from elevating cAMP levels: why so?

We employed CsChrimson, a red light-sensitive channelrhodopsin, to investigate the effect of optogenetic activation of each pC1 subset on EHP. Optogenetic activation of pC1a, pC1d, or pC1e had little effect on EHP; however, optogenetic activation of pC1b, c significantly increased EHP. This observation was puzzling because optogenetic silencing of the same neurons also increased EHP. In this experiment, females expressing CsChrimson were exposed to red light for the entire period of EHP measurement. Therefore, we suspect that prolonged activation of pC1b and pC1c neurons depleted their neurotransmitter pool, resulting in a silencing effect, but this requires further testing.

**Author response image 1. sa3fig1:** The prolonged optogenetic activation of pC1b, c neurons increases EHP, mimicking silencing of pC1b, c neurons. Females of the indicated genotypes were cultured on food with or without all-*trans*-retinal (ATR). The ΔEHP is calculated by subtracting the mean of the reference EHP of females cultured in control ATR- food from the EHP of individual females in comparison. The female genotypes are as follows: (A) *71G01-GAL4/UAS-CsChrimson*, (B) *pC1a-split-Gal4/UAS-CsChrimson*, (C) *pC1b,c-split-Gal4/UAS-CsChrimson*, (D) *pC1d-split-Gal4/UAS-CsChrimson*, and (E) *pC1e-split-Gal4/UAS-CsChrimson.* Gray circles indicate the ΔEHP of individual females, and the mean ± SEM of data is presented. Mann-Whitney Test (n.s. *p* > 0.05; **p* <0.05; *****p* < 0.0001). Numbers below the horizontal bar represent the mean of the EHP differences between the indicated treatments.

(4) Text editsIn general, the manuscript is very well-written, clear, and easy to follow. I recommend small edits of the text and correction of typos in some places:l.92: "*Drosophila* females seem to signal the social sexual context through sperm ejection." This sentence could give the impression that the main function of sperm ejection was to signal to conspecifics. I recommend reformulating to leave it open if ejected sperm is a signal or rather a simple cue. e.g. :"There is evidence that *Drosophila* females detect the social sexual context through sperm ejected by other females."

Thanks for the good suggestion. It has been revised as suggested. In addition, we have also made additional changes to the text to correct typos.

l.97: "transcriptional factor" > "transcription factor"

Revised as suggested. See lines 77, 98, and 201.

l.101: "There are Dsx positive 14 pC1 neurons in each brain hemisphere of the brain," > "There are 14 Dsx positive pC1 neurons in each brain hemisphere,"

Revised as suggested, it now reads " There are 14 Dsx-positive pC1 neurons in each hemisphere of the brain, ...".

l.160: ", even up to 1440 ng" > ", even when applied at concentrations as high as 1440 ng"

Revised as suggested.

l.168: "females with male oenocytes significantly shortens EHP" >"females with male oenocytes significantly shorten EHP"

Revised as suggested.

l.181: "it was restored when Orco expression is reinstated" >"it was restored when Orco expression was reinstated"

Revised as suggested. See line 186.

l.196: "MIES is almost completely abolished" >"MIES was almost completely abolished"

Revised as suggested. See line 201.

l.202: "a sexually dimorphic transcriptional factor gene" >"the sexually determination transcription factor gene" or "the sex specifically spliced transcription factor gene". The gene itself is not dimorphic!

Revised as suggested, lines 208-210 now read "The same study found that Dh44 receptor neurons involved in EHP regulation also express *doublesex* (*dsx*), which encodes sexually dimorphic transcription factors."

l.211: "to silenced" > "to silence"

Revised as suggested. See line 216.

l.229: "females that selectively produce the CRE-Luciferase reporter gene" >"females that selectively express CRE-Luciferase reporter"

Revised as suggested. See line 234.

l.271: "neurons. expedite" > delete dot

Revised as suggested. See line 284.

l.287: "Furthermore, our study has uncovered the conserved neural circuitry that processes male courtship cues and governs mating decisions play an important role in regulating this behavior." > grammar: "our study has uncovered that the conserved neural circuitry that processes male courtship cues and governs mating decisions plays an important role in regulating this behavior." Also: the meaning of "conserved" is not fully clear to me here: conserved in regards to other *Drosophila* species? Or do the authors mean: general functional similarity with mouse sexual circuitry?

The sentence (lines 299-301) has been revised for clarity to read "In addition, our study has revealed that the neural circuit that processes male courtship cues and controls mating decisions plays an important role in regulating this behavior. This fly circuit has recently been proposed to be homologous to VMHvl in the mouse brain (45, 46).”

l.311: "lipid drolet" > "lipid droplets"

Revised as suggested. See line 325.

l.316 and in several instances in the following, including Figure 5 caption (l.723) : "cAMP activity" > "cAMP levels" or "increased cAMP levels"

Revised as suggested.

l.323: "in hemibrain" > ", as seen in the hemibrain connectome dataset"

Revised as suggested. See line 337.

l.326: "increased cAMP levels causes pC1b,c neurons" > "increased cAMP levels cause pC1b,c neurons"

Revised as suggested. See line 340.

l.329: "removement" > "removal" or "ejection"

Revised as suggested, it now reads "the removal of the mating plug". See line 343.

l. 330: "This observation well aligns" > "The observation aligns well"

Revised as suggested. See line 345.

l. 398: Behavior assays: It would be good to describe how mating plug ejection was identified- by eye? Under the microscope/UV light?

The following sentence has been added to the behavioral assays section at lines 425-426: The sperm ejection scene, in which the female expels a white sac containing sperm and the mating plug through the vulva, has been directly observed by eye in recorded video footage.

l.685, Figure legend 2: "thermal activation" > "thermogenetic activation"

Revised as suggested. See line 430.

Reference:

(1) Doubovetzky, N., Kohlmeier, P., Bal, S., & Billeter, J. C. (2023). Cryptic female choice in response to male pheromones in *Drosophila melanogaster*. *bioRxiv*, 2023-12.